# Discovery and characterization of a prevalent human gut bacterial enzyme sufficient for the inactivation of a family of plant toxins

Nitzan Koppel[1], Jordan E Bisanz[2], Maria-Eirini Pandelia[3], Peter J Turnbaugh[2,4]*, Emily P Balskus[1,5]*

[1]Department of Chemistry and Chemical Biology, Harvard University, Cambridge, United States; [2]Department of Microbiology & Immunology, University of California, San Francisco, United States; [3]Department of Biochemistry, Brandeis University, Waltham, United States; [4]Chan Zuckerberg Biohub, San Francisco, United States; [5]Broad Institute, Cambridge, United States

**Abstract** Although the human gut microbiome plays a prominent role in xenobiotic transformation, most of the genes and enzymes responsible for this metabolism are unknown. Recently, we linked the two-gene 'cardiac glycoside reductase' (*cgr*) operon encoded by the gut Actinobacterium *Eggerthella lenta* to inactivation of the cardiac medication and plant natural product digoxin. Here, we compared the genomes of 25 *E. lenta* strains and close relatives, revealing an expanded 8-gene *cgr*-associated gene cluster present in all digoxin metabolizers and absent in non-metabolizers. Using heterologous expression and in vitro biochemical characterization, we discovered that a single flavin- and [4Fe-4S] cluster-dependent reductase, Cgr2, is sufficient for digoxin inactivation. Unexpectedly, Cgr2 displayed strict specificity for digoxin and other cardenolides. Quantification of *cgr2* in gut microbiomes revealed that this gene is widespread and conserved in the human population. Together, these results demonstrate that human-associated gut bacteria maintain specialized enzymes that protect against ingested plant toxins.

DOI: https://doi.org/10.7554/eLife.33953.001

*For correspondence:
Peter.Turnbaugh@ucsf.edu (PJT);
balskus@chemistry.harvard.edu (EPB)

## Introduction

The human gut microbiome extends the metabolic capabilities of the human body, extracting energy from otherwise indigestible dietary polysaccharides, synthesizing essential vitamins and amino acids, and modifying endogenous compounds. This microbial community also extensively metabolizes xenobiotics, including synthetic and natural product-based drugs, food additives, and environmental toxins (*Koppel et al., 2017*; *Spanogiannopoulos et al., 2016*). Previous work has focused on cataloguing the xenobiotics subject to transformation by gut microbes, their downstream products, and the extensive inter-individual variation in these activities (*Koppel et al., 2017*; *Spanogiannopoulos et al., 2016*). However, reduction of these complex metabolic networks to mechanism has been limited by our lack of knowledge about the microbial enzymes responsible for xenobiotic biotransformation.

Digoxin, a cardenolide used to treat heart failure and arrhythmia, represents a valuable test case for our ability to elucidate the biochemical and evolutionary underpinnings of gut microbial xenobiotic metabolism. It has been known for decades that human gut bacteria reduce digoxin to the inactive metabolite dihydrodigoxin, decreasing drug efficacy and toxicity (*Saha et al., 1983*;

**eLife digest** Trillions of microbes live within the human gut and influence our health. In particular, these microbes can modify food and drugs into compounds (metabolites) that humans cannot produce on their own. These compounds are often beneficial to the human host, but in some cases – for example, if the modification alters how a drug works – can be detrimental.

Digoxin is a toxic chemical produced by plants that, in low doses, can be used to treat heart conditions. It has been known for decades that the human gut bacterium *Eggerthella lenta* transforms digoxin into a metabolite that is an ineffective drug. Microbes use biological catalysts called enzymes to produce metabolites, but it was not known which enzymes enable *E. lenta* to modify digoxin.

Using biochemical and genomic techniques, Koppel et al. now show that an enzyme called Cgr2 inactivates digoxin and other related plant toxins. Data about the gut microbes in nearly 1,900 people from three continents revealed that bacteria that can produce Cgr2 were present in the guts of more than 40% of the individuals, although often in low abundance. Further experiments did not reveal any obvious benefits that *E. lenta* gains from modifying digoxin. Instead, Koppel et al. propose that the bacteria carry out this modification to protect their human host from plant toxins.

The results presented by Koppel et al. emphasise that the activities of gut microbes should be considered when designing new drugs or assessing how they work in the human body. The strategies used to identify Cgr2 could now be applied to discover other important gut microbe-drug interactions. Ultimately, this knowledge will help us to predict and control the activities of gut microbes in ways that could improve human health.
DOI: https://doi.org/10.7554/eLife.33953.002

*Lindenbaum et al., 1981a*; *Lindenbaum et al., 1981b*). Screening hundreds of gut bacterial strains from humans that excreted high levels of dihydrodigoxin revealed only two isolates that were capable of metabolizing digoxin, both of which were strains of the anaerobic, low abundance bacterium *Eggerthella lenta* (*Saha et al., 1983*). However, the presence of *E. lenta* in the gut microbiome cannot accurately predict this reactivity, as patients colonized by *E. lenta* still show marked variation in dihydrodigoxin production (*Mathan et al., 1989*; *Alam et al., 1988*). We identified two mechanisms underlying this discrepancy: strain-level variations in the *E. lenta* population and inhibition of bacterial drug metabolism by dietary amino acids. Digoxin induces expression of a 2-gene operon encoded by the type strain of *E. lenta* (DSM 2243), which we named the cardiac glycoside reductase (*cgr*) operon (*Haiser et al., 2013*). The *cgr* operon was absent in two *E. lenta* strains that did not metabolize digoxin ('non-reducing' strains) and *cgr* operon presence and abundance predicted the extent of drug inactivation by human gut microbial communities in ex vivo incubations (*Haiser et al., 2013*). Furthermore, germ-free mice that had been mono-colonized by a reducing (*cgr+*) strain of *E. lenta* had lower serum levels of digoxin than mice colonized by a non-reducing (*cgr-*) strain, and dietary arginine efficiently blocked digoxin reduction by the *cgr* operon in *cgr⁺ E. lenta*-colonized mice (*Haiser et al., 2013*; *Haiser et al., 2014*).

Although it is tempting to consider applying these insights to develop novel microbiome-based diagnostics and co-therapies (*Spanogiannopoulos et al., 2016*), multiple critical questions remained unaddressed. Our original studies were entirely based on the *E. lenta* type strain, isolated in 1938 from a rectal cancer biopsy (*Moore et al., 1971*); thus, the presence of *cgr +E. lenta* in the modern-day human gastrointestinal tract was unclear. Although we had associated the *cgr* operon with digoxin reduction, the minimal genetic machinery necessary and sufficient for this biotransformation had not been determined. Perhaps most importantly, the specificity of the digoxin-reducing enzyme (s) for cardenolides and their ability to accept additional endogenous or ingested substrates remained unclear.

Here, we used a combination of comparative genomics, heterologous expression, biochemistry, and metagenomics to address these long-standing questions. We uncovered a highly conserved cluster of genes that co-occurs with the *cgr* operon, representing a single genetic locus predictive of digoxin metabolism. We demonstrated that a single protein encoded by this locus, Cgr2, is sufficient for digoxin reduction and is widespread in human gut microbiomes. Cgr2 is a novel oxygen-sensitive

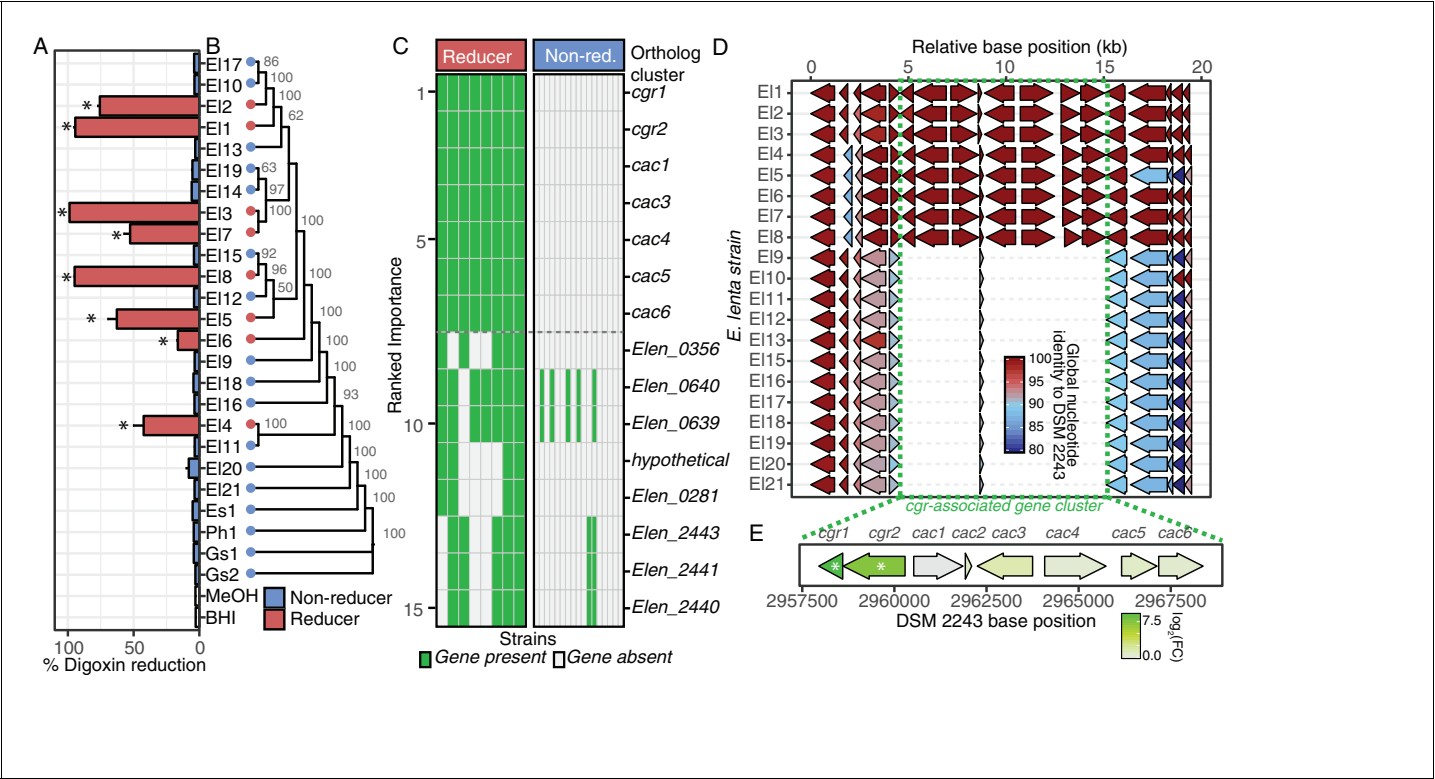

**Figure 1.** Comparative genomics expands the boundaries of the *cgr* operon. (A) Survey of digoxin reduction in 21 strains of *E. lenta* (El#), 2 strains of *Gordonibacter* spp. (Gs#), *E. sinensis* (Es1), and *Paraeggerthella hongkongesis* (Ph1) (*Figure 1—source data 1* and *2*) revealed eight strains capable of reducing digoxin to dihydrodigoxin (*p<0.05, ANOVA with Dunnett's test vs. vehicle controls). Data represents mean ± standard error of the mean (SEM) over three biological replicates. (B) Digoxin reduction did not correlate with phylogeny in *E. lenta* species (cladogram displayed with bootstrap values indicated at nodes; p=0.275, K = 0.049, Blomberg's K). (C) Comparative genomics using a random forest classifier (*see Materials and methods*) revealed seven genes with perfect predictive accuracy for digoxin reduction. The orthologous cluster identified as hypothetical corresponds to an open reading frame present at position 299442..2995131 in the DSM 2243 reference genome. (D) Analysis of genomic context revealed a highly conserved 10.4 kb locus of 7 genes that flank a short, conserved hypothetical gene, herein termed the *cgr*-associated gene cluster (*cac*). (E) Analysis of gene expression in the *cgr*-associated gene cluster revealed only the *cgr*-locus was significantly upregulated by exposure to digoxin. * FDR < 0.1 (*Figure 1— source data 3*).

DOI: https://doi.org/10.7554/eLife.33953.003

The following source data is available for figure 1:

**Source data 1.** Bacterial strains used in study.
DOI: https://doi.org/10.7554/eLife.33953.004
**Source data 2.** Digoxin Reduction by *E.lenta* and related bacterial isolates.
DOI: https://doi.org/10.7554/eLife.33953.005
**Source data 3.** Differential gene expression analysis of *cgr*-associated genes in *E.lenta* DSM 2243.
DOI: https://doi.org/10.7554/eLife.33953.006

reductase that requires flavin adenine dinucleotide (FAD) and at least one [4Fe-4S] cluster. Surprisingly, we found that Cgr2 only accepts digoxin and other cardenolides, prompting the provocative hypothesis that the gut microbiome provides a first-line of protection against ingested toxins analogous to that of host enzymes expressed in the intestinal epithelium and liver. Finally, this work establishes a generalizable framework for mechanistic investigations of gut microbial xenobiotic metabolism that will enhance our understanding of the complex dietary, host, and microbial factors that impact pharmacology and toxicology, and provide a stronger foundation for translational studies in patient populations.

## Results

### Identification of a single genetic locus conserved in all digoxin-reducing strains

*E. lenta* strains vary in their ability to reduce digoxin (*Haiser et al., 2013*; *Haiser et al., 2014*); however, our prior attempts at identifying the minimal genetic machinery necessary for metabolism were limited by the availability of just a single strain capable of this activity (*E. lenta* DSM 2243). Through public repositories and isolation of novel strains, we curated, sequenced, and annotated a collection of 25 *E. lenta* and closely related *Coriobacteriia* strains (*Bisanz et al., 2018*). These bacteria were isolated from 22 individuals in 6 countries across three continents spanning the years of 1938–2015 (*Figure 1—source data 1*).

We used liquid chromatography-tandem mass spectrometry (LC-MS/MS) to quantify the biotransformation of digoxin to dihydrodigoxin by each strain and identified seven additional strains capable of drug inactivation (*Figure 1A*). Culturing experiments were performed using 10 µM of digoxin, which is within the estimated range (0.4–14.6 µM) of therapeutic concentrations of the drug in the gut prior to absorption by host epithelial cells (*Schiller et al., 2005*). The digoxin metabolizing strains did not exhibit a significant phylogenetic signal (*Figure 1B*; p=0.275, K = 0.049, Blomberg's K test), suggesting that this phenotypic trait has been gained (or lost) multiple times over the course of *E. lenta* evolution. Machine learning (random forests) analysis of orthologous gene cluster presence/absence across the strain collection revealed a single genetic locus with 100% discriminative value between metabolizers and non-metabolizers (*Figure 1C*). This locus, referred to hereafter as the *cgr* gene cluster, includes the previously identified 2-gene *cgr* operon (*cgr1* and *cgr2*) and six neighboring genes termed *cac* (*cgr*-associated cluster) genes (*Figure 1D*). The *cac* genes include a putative LuxR type transcriptional regulator (Cac3), a predicted flavin-dependent fumarate reductase (Cac4), three proteins of unknown function (Cac1, Cac5, Cac6), and a short protein (Cac2) that is conserved in both *cgr-* and *cgr+* strains of *E. lenta*.

The 10.4 kb *cgr* gene cluster was highly conserved between strains with an average global nucleotide identity of 99.95 ± 0.05% (mean ± standard deviation (SD)). Both *cgr+* and *cgr-* strains share a short 174 bp hypothetical gene (*cac2*) that is conserved with 100% global nucleotide identity in *cgr + strains*, while *cgr-* strains are 90.20–91.37% identical to *cgr+ cac2*. This conservation and genomic context may be indicative of multiple translocations in the region creating the *cgr*-associated gene cluster although obvious markers of recent translocation of the *cgr* gene cluster are absent.

### Cgr2 is sufficient for digoxin reduction

Multiple lines of evidence suggested that the *cgr* operon encodes the enzymes responsible for digoxin metabolism. Of the eight genes in the *cgr* gene cluster, only three show primary sequence homology to reductases: *cgr1*, *cgr2*, and *cac4*. RNA sequencing (*Haiser et al., 2013*) demonstrated that the *cgr* operon (*cgr1* and *cgr2*) is highly upregulated (>165 fold) in response to digoxin, whereas *cac4* is not significantly induced (1.3-fold change relative to vehicle controls, p=0.83) (*Figure 1E*). The remainder of the *cgr*-associated cluster is largely transcriptionally dormant during exponential growth both with and without the presence of digoxin (<6 normalized counts per gene) and is therefore unlikely to be linked to digoxin metabolism. Initial annotations of Cgr1 and Cgr2 suggested both proteins might mediate digoxin reduction. Cgr1 is a putative membrane-anchored protein that belongs to the cytochrome c3 superfamily (Pfam 14537) and is predicted to harbor covalently bound heme groups (CXXCH motif). It most closely resembles the NapC/NirT (NrfH) family of proteins that transfer electrons from the membrane quinone pool to associated reductases, facilitating reduction of terminal electron acceptors such as nitrite and sulfite (*Kemp et al., 2010*; *Kern et al., 2008*). We also identified a close homolog of Cgr1 (Elen_2528) in *E. lenta* DSM 2243 (91.75% amino acid identity, BLASTP) that is a component of the *E. lenta* core genome (99.39 ± 0.81% global identity mean ± SD). The presence of this highly similar protein in both metabolizing and non-metabolizing strains further indicates that Cgr1 is involved in a more general function (*e.g.* electron transfer, membrane docking) rather than direct reduction of digoxin. On the other hand, Cgr2 is unique to the genomes of *cgr+ E. lenta*, and the closest homologs of Cgr2 display <28% amino acid identity. Cgr2 is a homolog of flavin adenine dinucleotide (FAD)-dependent fumarate reductases (Pfam 00890; Interpro 003953/027477) and is predicted to undergo secretion via the twin

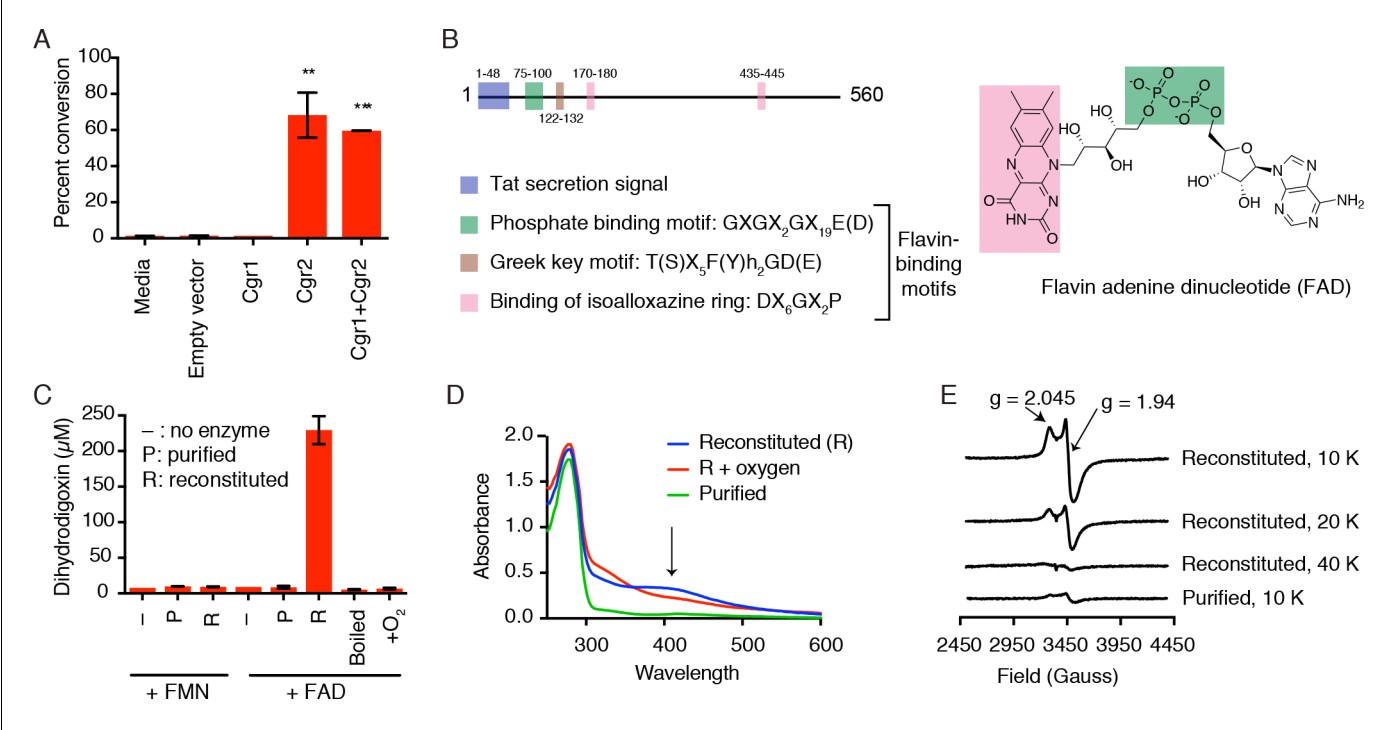

**Figure 2.** Cgr2 is sufficient for digoxin reduction and requires FAD and [4Fe-4S] cluster(s) for activity. (**A**) Whole cell assays using *R. erythropolis* expressing Cgr1 and Cgr2 constructs demonstrated that Cgr2 is sufficient for reducing digoxin. Data represents the mean ± SEM (n = 3 biological replicates). Asterisks indicate statistical significance of each variant as compared to empty vector by Student's t test (**p<0.01, ***p<0.001) (*Figure 2— source data 1*). (**B**) Annotation and amino acid numbering of Cgr2, including the predicted Tat secretion signal and three conserved flavin-binding motifs from the glutathione reductase family (X = any amino acid; h = hydrophobic residue). (**C**) In vitro activity of Cgr2 for digoxin reduction using reduced methyl viologen as an electron donor, analyzed by liquid chromatography-tandem mass spectrometry (LC-MS/MS). [Fe-S] cluster reconstitution, FAD, and anaerobic conditions are required for Cgr2 activity. Data represents the mean ± SEM (n = 3 independent experiments) (*Figure 2—source data 2*). FAD = flavin adenine dinucleotide; FMN = flavin mononucleotide. (**D**) Ultraviolet-visible (UV-Vis) absorption spectra of Cgr2 revealed an oxygen-sensitive peak centered around 400 nm that increased upon [Fe-S] cluster reconstitution, supporting the presence of [4Fe-4S] clusters in Cgr2. (**E**) Electron paramagnetic resonance (EPR) spectra of sodium dithionite-reduced Cgr2 reconstituted with iron ammonium sulfate hexahydrate ((NH$_4$)$_2$Fe(SO$_4$)$_2$·6H$_2$O) and sodium sulfide (Na$_2$S·9H$_2$O). G-values and decreased EPR signal intensity at higher temperatures (10 – 40 K) indicated the presence of low potential [4Fe-4S]$^{1+}$ clusters. Experimental conditions were microwave frequency 9.38 GHz, microwave power 0.2 mW, modulation amplitude 0.6 mT, and receiver gain 40 dB.

DOI: https://doi.org/10.7554/eLife.33953.007

The following source data and figure supplements are available for figure 2:

**Source data 1.** Digoxin metabolism by *R. erythropolis* overexpressing Cgr proteins.
DOI: https://doi.org/10.7554/eLife.33953.011

**Source data 2.** Digoxin metabolism by Cgr2 in vitro.
DOI: https://doi.org/10.7554/eLife.33953.012

**Source data 3.** Examples of [2Fe-2S], [3Fe-4S], and [4Fe-4S] cluster binding motifs that are not found in Cgr2 (*Zhang et al., 2010*; *Nakamaru-Ogiso et al., 2002*; *Lee et al., 2004*; *Pandelia et al., 2011*; *Schnackerz et al., 2004*; *Leech et al., 2003*; *Gorodetsky et al., 2008*; *Lee et al., 2010*; *Weiner et al., 2007*; *Klinge et al., 2007*; *Dickert et al., 2002*; *Conover et al., 1990*; *Schneider and Schmidt, 2005*; *Iwasaki et al., 2000*; *Banci et al., 2013*; *Dailey and Dailey, 2002*; *Jung et al., 2000*).
DOI: https://doi.org/10.7554/eLife.33953.013

**Source data 4.** Digoxin metabolism by *R. erythropolis* overexpressing Cgr2 cysteine to alanine point mutants.
DOI: https://doi.org/10.7554/eLife.33953.014

**Source data 5.** Digoxin metabolism by Cgr2 cysteine to alanine point mutants in vitro.
DOI: https://doi.org/10.7554/eLife.33953.015

**Figure supplement 1.** [Fe-S] cluster(s) affect Cgr2 stability and oligomerization.
DOI: https://doi.org/10.7554/eLife.33953.008

**Figure supplement 2.** Cgr2 activity, but not EPR-active [4Fe-4S] clusters, increase with higher Fe and S equivalents.
DOI: https://doi.org/10.7554/eLife.33953.009

**Figure supplement 3.** Identification of 6 cysteine residues important for Cgr2 activity.

*Figure 2 continued*

DOI: https://doi.org/10.7554/eLife.33953.010

arginine translocation (Tat) pathway. Taken together, these observations led us to hypothesize that Cgr1 and Cgr2 form a membrane-associated complex that catalyzes reduction of the α,β-unsaturated butyrolactone of digoxin.

Heterologous expression of Cgr1 and Cgr2 in the model Actinobacterium *Rhodococcus erythropolis* L88 (*Mitani et al., 2005*; *Nakashima and Tamura, 2004a*, *2004b*) allowed us to test whether these proteins were sufficient for digoxin reduction. After inducing protein expression, cultures were incubated with 10 μM of digoxin and dihydrodigoxin production was quantified by LC-MS/MS (*Figure 2A*). The Cgr2 expressing strains showed a significant increase in dihydrodigoxin levels relative to empty vector controls (*Figure 2A*; *Figure 2—source data 1*). In contrast, no activity was observed for the strain expressing only Cgr1, although this could be due to a lack of protein, as no overexpression of Cgr1 could be detected in either clarified lysates or membrane fractions. These results show that Cgr2 is sufficient for digoxin reduction in *R. erythropolis* cells, and endogenous redox active proteins and/or metabolites in this heterologous host may fulfill the putative function of Cgr1 as an electron donor.

## Cgr2 activity depends on [4Fe-4S] cluster(s)

Having identified Cgr2 as the critical reductase enzyme, we next aimed to reconstitute its activity in vitro. Examining multiple tagged versions and truncations of Cgr2 in *R. erythropolis* revealed that a Cgr2(–48aa)-NHis$_6$-tagged construct lacking the Tat secretion signal gave the highest yield and purity (*Figure 2—figure supplement 1A*). This construct, hereafter referred to as 'wild-type' Cgr2, was used for all in vitro studies. Computational analysis of Cgr2 predicted that it would bind flavin through a Rossmann fold (*Dym and Eisenberg, 2001*), and the three motifs required for cofactor binding are conserved in all 8 Cgr2 sequences (*Figure 2B*). However, Cgr2 did not co-purify with flavin. Moreover, Cgr2 obtained from initial purifications was thermally unstable (melting temperature <37°C), was prone to degradation during cell lysis (*Figure 2—figure supplement 1*), and displayed low activity for digoxin reduction (*Figure 2C*; *Figure 2—source data 2*). Together, these observations indicated that an essential cofactor was likely missing. We also noticed that purified Cgr2 was light brown in color, suggesting the presence of a metallocofactor. Certain flavin-dependent reductases use metallocofactors to mediate the transfer of electrons to the active site, including cytochromes *c* in soluble enzymes and oxygen-sensitive iron-sulfur ([Fe-S]) clusters in membrane-bound enzymes (*Kern et al., 2008*; *Iverson et al., 2002*; *Leys et al., 1999*). The combination of the brown color and the presence of 16 cysteines in the mature Cgr2 sequence led us to hypothesize that this enzyme contained one or more [Fe-S] clusters. However, we were unable to detect any canonical [2Fe-2S], [3Fe-4Fe], or [4Fe-4S] cluster binding motifs within the Cgr2 sequence (*Figure 2—source data 3*) (*Zhang et al., 2010*; *Nakamaru-Ogiso et al., 2002*; *Lee et al., 2004*; *Pandelia et al., 2011*; *Schnackerz et al., 2004*; *Leech et al., 2003*; *Gorodetsky et al., 2008*; *Lee et al., 2010*; *Weiner et al., 2007*; *Klinge et al., 2007*; *Dickert et al., 2002*; *Conover et al., 1990*; *Schneider and Schmidt, 2005*; *Iwasaki et al., 2000*; *Banci et al., 2013*; *Dailey and Dailey, 2002*; *Jung et al., 2000*).

We therefore sought to determine whether Cgr2 required an [Fe-S] cluster to catalyze digoxin reduction. Attempts to chemically reconstitute [Fe-S] cluster formation by incubating Cgr2 with iron and sulfide under anaerobic conditions greatly improved protein stability (*Figure 2—figure supplement 1B–D*). In addition to performing this reconstitution step, including FAD in assay mixtures dramatically increased the digoxin reduction activity of purified protein (*Figure 2C*). Even after reconstitution, Cgr2 required excess FAD for maximal activity, suggesting that Cgr1 could be important for enhancing FAD binding, as has been observed for proteins whose FAD-binding site is predicted to occur at the interface of two domains (*Kleven et al., 2015*).

Having demonstrated that [Fe-S] reconstitution was essential for activity, we next attempted to determine the exact nature of this metallocofactor. Prior to reconstitution, purified Cgr2 contained between 0.2–0.6 equivalents of iron and sulfide, and its ultraviolet-visible (UV-vis) spectrum displayed absorption features consistent with the presence of low levels of [Fe-S] clusters (*Ayala-*

*Castro et al., 2008*) (*Figure 2D*). After reconstitution, Cgr2 exhibited a broad peak around 400 nm (*Figure 2D*) that decreased in absorbance upon addition of an excess amount of the reducing agent sodium dithionite (*Figure 2—figure supplement 2A*). These spectral properties are characteristic of redox-active [Fe-S] clusters. Exposure of reconstituted Cgr2 to oxygen led to [Fe-S] cluster decomposition as evidenced by a decrease in the absorbance at 400 nm (*Figure 2D*), demonstrating that the [Fe-S] cluster(s) of Cgr2 are oxygen-sensitive.

Although these UV-vis experiments indicated the presence of [Fe-S] cluster(s) in Cgr2, they could not define the precise structures of these cofactors. To more definitively characterize these metallocofactors, we turned to electron paramagnetic resonance (EPR) spectroscopy. This technique detects unpaired electrons and can differentiate between the various types of [Fe-S] clusters as well as provide information about cluster orientation and redox state. In the absence of a reducing agent, purified, unreconstituted Cgr2 does not have an EPR signal (*Figure 2—figure supplement 2B*). Upon reduction with sodium dithionite, the EPR spectrum of Cgr2 exhibits a signal with axial symmetry and with principal g-components of 2.045 and 1.94. This signal increases in intensity upon reconstitution of Cgr2 (10 K), and is barely detectable above 40 K (*Figure 2E*). Both the principal g-values and the relaxation properties (temperature dependence) of this signal are characteristic of low-potential tetranuclear $[4Fe-4S]^{1+}$ centers. These observations indicate that Cgr2 contains $[4Fe-4S]^{2+}$ cluster that can undergo reduction to the corresponding $[4Fe-4S]^{1+}$ state. These redox properties suggest that these [4Fe-4S] cluster(s) may participate in catalysis (electron transfer).

Next, we sought to determine how many [4Fe-4S] clusters were present in Cgr2. Using a $Cu^{2+}$-EDTA standard, we determined that purified, unreconstituted Cgr2 contains 0.02–0.03 $[4Fe-4S]^{1+}$ clusters per protein monomer. After reconstitution with iron and sulfide, the intensity of the EPR signal increased to 0.13–0.25 $[4Fe-4S]^{1+}$ clusters per Cgr2. Though these data may suggest the presence of one [4Fe-4S] cofactor per Cgr2, they do not exclude the possibility of multiple [Fe-S] centers. Indeed, the in vitro activity of Cgr2 increases upon reconstitution with increasing equivalents of iron and sulfide (*Figure 2—figure supplement 2C–D*). Additional spectroscopic or structural characterization (*e.g.* crystallography) will be required to definitively determine the number of [Fe-S] cluster(s) present in Cgr2.

## Identification of amino acids required for Cgr2 function

As motif analysis could not identify putative [4Fe-4S] cluster binding sites in Cgr2, we attempted to use site-directed mutagenesis to reveal the cysteine residues required for cofactor assembly. Individually mutating each of the 16 cysteines present in wild-type Cgr2 to alanine revealed six residues that, when mutated, significantly decreased dihydrodigoxin production by both heterologously expressed and purified Cgr2 (*Figure 2—figure supplement 1A*; *Figure 2—figure supplement 3A–C*; *Figure 2—source data 4*; *Figure 2—source data 5*). EPR analysis of these six Cgr2 mutants revealed comparable levels of $[4Fe-4S]^{1+}$ cluster incorporation relative to the wild-type enzyme (*Figure 2—figure supplement 3D*), which may argue against the involvement of these cysteines in [4Fe-4S] cluster ligation. However, substitution of a single cysteine residue may not always be sufficient to prevent [4Fe-4S] cluster formation (*Iismaa et al., 1991*; *Hewitson et al., 2002*; *Martín et al., 1990*).

Alternatively, these six cysteines may be critical for protein structure (*e.g.* through participating in disulfide formation) or could coordinate another metal center not detectable in our spectroscopic experiments. Consistent with this latter proposal, we found that a range of divalent metal cations ($Fe^{2+}$, $Mn^{2+}$, $Mg^{2+}$) stimulated the activity of Cgr2 in vitro (*Figure 2—figure supplement 3E*) without altering protein stability. Additionally, $Fe^{2+}$ stimulated the in vitro activity of only 3 out of 6 impaired mutants (C158A, C187A, C327A) (*Figure 2—figure supplement 3F*). Notably, binding of digoxin to its target in human cells, $Na^+/K^+$ ATPase, is thought to be mediated by long-range electrostatic interactions between a $Mg^{2+}$ ion and the electron rich, partially negatively charged oxygen atom of the unsaturated lactone (*Laursen et al., 2015*; *Weigand et al., 2014*). It is possible that the three remaining cysteine residues (C82, C265, C535) could influence binding of a divalent metal cation that similarly positions or activates digoxin in the Cgr2 active site.

To identify additional amino acids that may be important for Cgr2 function, we compared the Cgr2 sequences encoded within our collection of *E. lenta* genomes. Strikingly, only two *cgr2* nucleotide variants were detected, which were validated by targeted Sanger sequencing. One of these variants is only found in the DSM 2243 type strain resulting in a conservative methionine (M) to valine (V) substitution at position 381. The other results in either aromatic tyrosine (Y) as in the type strain

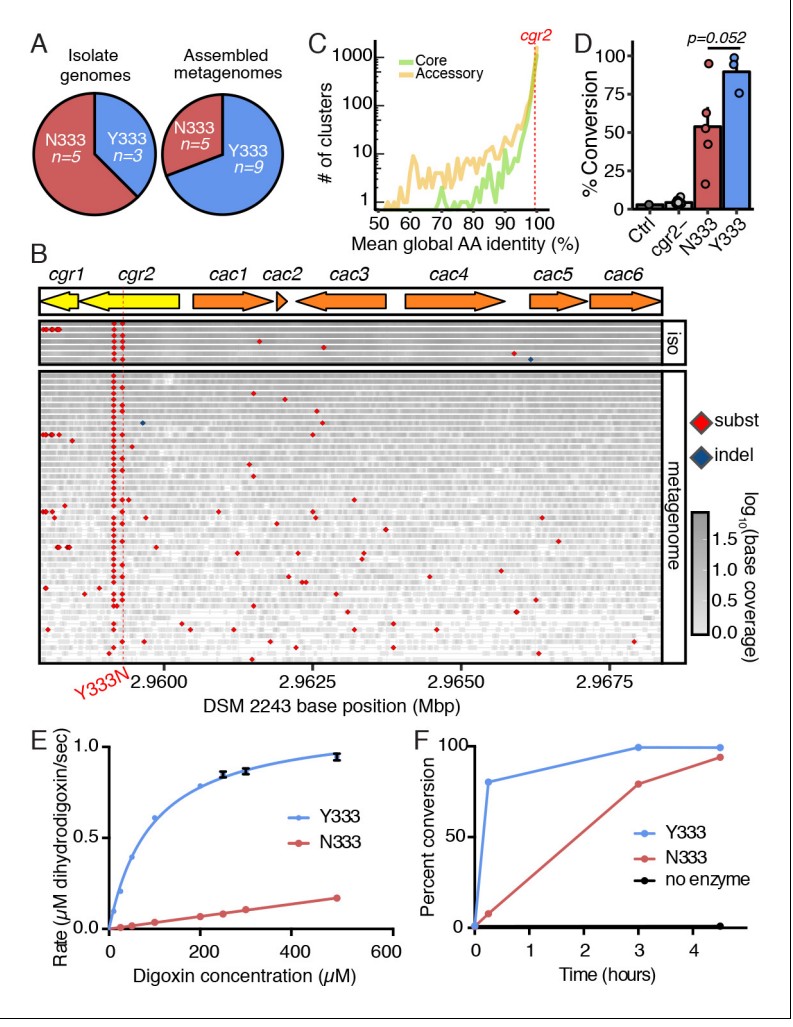

**Figure 3.** A single polymorphism in Cgr2 at position 333 (Y/N) leads to altered metabolism of digoxin. (**A**) Analysis of Cgr2 amino acid sequence composition of isolate genomes (n = 8) and reconstructed sequences from gut microbiome datasets (n = 14) revealed a single non-conservative Y333N variant in isolate strains supported by metagenomes (12Y/10N). (**B**) Nucleotide variation in the *cgr*-associated gene cluster. Reads aligned from both isolate genomes (iso) and metagenomes were aligned to the DSM 2243 reference assembly and plotted if there was coverage of the Y333N variant position (CP001726.1: 2959294 bp). Variants were called when at least one read was mapped to the position and > 50% of reads supported an alternative base. Read depth at any given position is indicated by shading. Confirming assembly-based methods, *cgr2* amino acid position 333 was bi-allelic with 5 of 8 isolate genomes and 15 of 49 metagenomes showing the N333 variant (four metagenomes have evidence of both alleles) and minimal variation in other regions of the cluster. (**C**) Average amino acid conservation in the *E. lenta* core (n = 1832) and non-singleton accessory genome (n = 2557) demonstrates that *cgr2* is in the 67th percentile for conservation in the pan-genome (78.8[th] in the core genome, and 58.5[th] in the non-singleton accessory genome) with higher average conservation observed in the core genome (98.6 ± 2.41% core, 97.4 ± 5.6% accessory, mean ± SD). (**D**) Comparison of digoxin metabolism in culture by *E. lenta cgr2-* (n = 13 strains), Cgr2Y333 (n = 3 strains), and Cgr2N333 (n = 5 strains). Control refers to digoxin in BHI media. Each point represents the mean percent conversion to dihydrodigoxin of each individual strain cultured in biological triplicate. Bars represent the mean ± SEM percent conversion per *E. lenta* group. Statistical significance between Y333 and N333 groups was calculated using two-tailed Welch's t test (p=0.052) (*Figure 3—source data 1*). (**E**) Michaelis–Menten kinetics of Cgr2 towards digoxin revealed that the Y333 variant is significantly more active than the N333 variant. Data represents mean ± SEM (n = 3 independent experiments) (*Figure 3—source data 2*; *Figure 3—source data 4*). (**F**) In vitro time course (0 – 4.5 hr) of the conversion of digoxin to dihydrodigoxin by Cgr2 Y333 and N333 variants. Reaction aliquots were quenched in methanol and analyzed by liquid chromatography-tandem mass spectrometry. Values represent mean ± SEM (n = 3 independent experiments).
*Figure 3 continued on next page*

*Figure 3 continued*

Asterisks indicate statistical significance at each timepoint of Y333 vs. N333 percent conversion, by Student's t test (*p<0.05, ***p<0.001) (*Figure 3—source data 3*).
DOI: https://doi.org/10.7554/eLife.33953.016

The following source data and figure supplement are available for figure 3:

**Source data 1.** Whole cell activity of *E. lenta* strains with Y333 vs N333 Cgr2 variants.
DOI: https://doi.org/10.7554/eLife.33953.017
**Source data 2.** Kinetics of Y333 and N333 Cgr2 variants towards digoxin.
DOI: https://doi.org/10.7554/eLife.33953.019
**Source data 3.** In vitro time course of digoxin reduction by Y333 and N333 Cgr2 variants.
DOI: https://doi.org/10.7554/eLife.33953.020
**Source data 4.** Comparison of Cgr2 kinetics with related enzymes towards their respective substrates (*Kemp et al., 2010*; *Rohman et al., 2013*; *Morris et al., 1994*; *Bogachev et al., 2012*).
DOI: https://doi.org/10.7554/eLife.33953.021
**Figure supplement 1.** Phylogenetic tree of assayed *E. lenta* strains showing *cgr2* Y333/N333 variants.
DOI: https://doi.org/10.7554/eLife.33953.018

DSM 2243 or neutral asparagine (N) at position 333 (*Figure 3A*). We were also able to fully or partially reconstruct 14 additional *cgr2* sequences using reads mapping to the *cgr* gene cluster from 96 gut microbiome datasets with a high abundance of *E. lenta* (>1x coverage or >0.001 proportional abundance). These metagenome fragments confirmed the presence of both Y333 and N333 variants in a 9:5 ratio (*Figure 3A*) while the DSM 2243 M381 variant was not observed. To avoid biases against lower *E. lenta* coverage metagenomes, we also applied an assembly-free method based on calling variants from aligned reads (*Figure 3B*). This uncovered 49 metagenomes with at least one read mapping over the variant position, confirming the bi-allelic nature with 15 Y333 and 34 N333 metagenomes. Nearly all metagenomes (41/42) with reads mapping to position 381 supported the valine residue suggesting that the DSM 2243 M381 variant is rare. Given that this analysis confirmed the highly conserved nature of the *cgr* locus, we analyzed the conservation of *cgr2* in the context of the *E. lenta* pan-genome (based on 24 sequenced isolates) finding that it is at the 67th percentile of conservation. These results suggest that *cgr2* sequence conservation is not unusual for this species, with the caveat that relatively few genomes were available for analysis (*Figure 3C*).

Given the ubiquity of variation at position 333, we assessed its functional consequences by comparing the activity of the two Cgr2 variants in vivo and in vitro. *E. lenta* strains encoding the N333 variant show a trend towards a decreased ability to metabolize digoxin as compared to Y333-encoding strains (*Figure 3D*; *Figure 3—source data 1*; p=0.052 Welch's t-test). This decreased activity was more readily apparent following incubation of digoxin with Cgr2 proteins in vitro (*Figure 3E*; *Figure 3—source data 2*). While kinetic parameters for wild-type Cgr2 (Y333) were $K_M = 94.6 \pm 7.1$ μM and a catalytic efficiency of $2.4 \pm 0.8 \times 10^3$ $M^{-1} s^{-1}$, saturating $V_{max}$ conditions could not be reached for the N333 variant within the range of concentrations where digoxin is soluble ($\leq 0.5$ mM). Despite its lower activity, Cgr2 N333 converted digoxin to dihydrodigoxin at a comparable efficiency to Y333 after 4.5 hr (*Figure 3F*; *Figure 3—source data 3*). Compared to the activity of other FAD-dependent reductases towards their native substrates, the Y333 Cgr2 variant is less efficient for digoxin reduction (*Kemp et al., 2010*; *Rohman et al., 2013*; *Morris et al., 1994*; *Bogachev et al., 2012*) (*Figure 3—source data 4*). This decreased activity could arise from impaired cofactor binding or reconstitution, or inefficient electron transfer in vitro in the absence of Cgr1. Alternatively, these results could indicate that digoxin is not the endogenous substrate of Cgr2.

## Cgr2 is a novel enzyme that preferentially reduces cardenolides

To systematically test for additional Cgr2 substrates, we assessed the enzyme's activity toward 28 small molecules using a colorimetric assay (*Figure 4*; *Figure 4—figure supplement 1*). These molecules were selected based on their chemical similarity to digoxin and their relevance in the context of the human gut. Cgr2 displayed robust activity only toward cardenolides, the family of plant toxins that includes the pharmaceutical agents digoxin and digitoxin as well as ouabain, which is used as an arrowhead poison (*Michalak et al., 2017*). The cardenolide aglycones digoxigenin and ouabagenin were metabolized at a significantly faster rate than their glycosylated forms digoxin and ouabain,

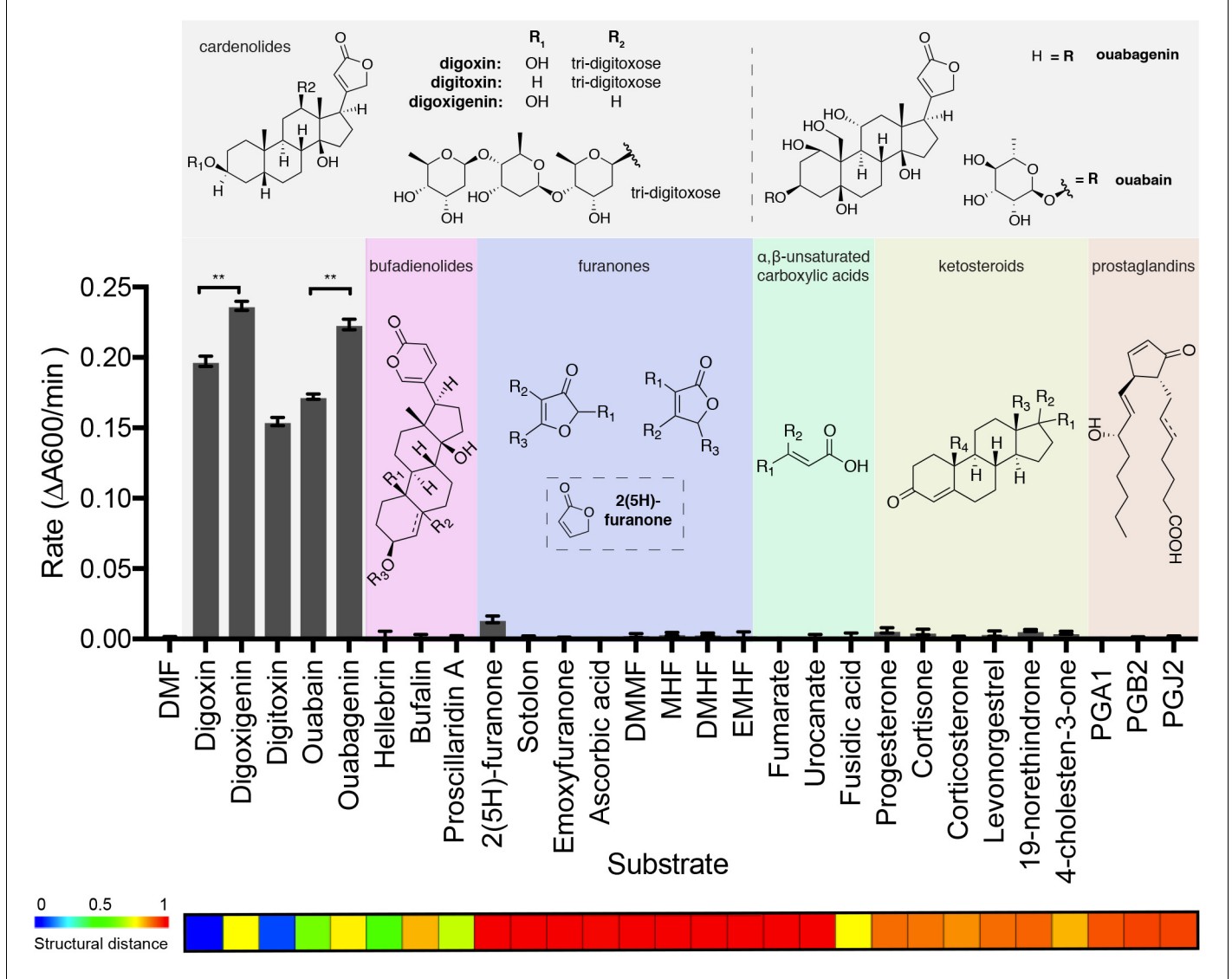

**Figure 4.** The substrate scope of Cgr2 is restricted to cardenolides. Rate of methyl viologen oxidation coupled to substrate reduction by Cgr2. Colors denote different substrate classes. With the exception of the cardenolides, a representative substrate structure is shown. Values represent mean ± SEM (n = 3 independent experiments). **p<0.01, Student's *t* test (*Figure 4—source data 1*). The heatmap generated in ChemMine (*Backman et al., 2011*) represents the structural similarity of each compound relative to digoxin. Structural distance matrix is calculated as (1- Tanimoto coefficient), where lower values represent more structurally similar compounds.

DOI: https://doi.org/10.7554/eLife.33953.022

The following source data and figure supplements are available for figure 4:

**Source data 1.** Rate of methyl viologen oxidation coupled to substrate reduction by Cgr2.
DOI: https://doi.org/10.7554/eLife.33953.025

**Figure supplement 1.** Putative substrates for Cgr2 in the context of the human gut.
DOI: https://doi.org/10.7554/eLife.33953.023

**Figure supplement 2.** Digoxin and related cardenolides do not affect *E. lenta* growth in rich or minimal media.
DOI: https://doi.org/10.7554/eLife.33953.024

respectively (**p<0.01, Student's *t* test). The isolated lactone 2(5H)-furanone was minimally processed by Cgr2, indicating that an intact steroid core is important for activity. However, the qualitatively similar rates observed for reduction of the various cardenolides demonstrates that the enzyme tolerates differences in the number and position of hydroxyl groups on the steroid scaffold. Overall,

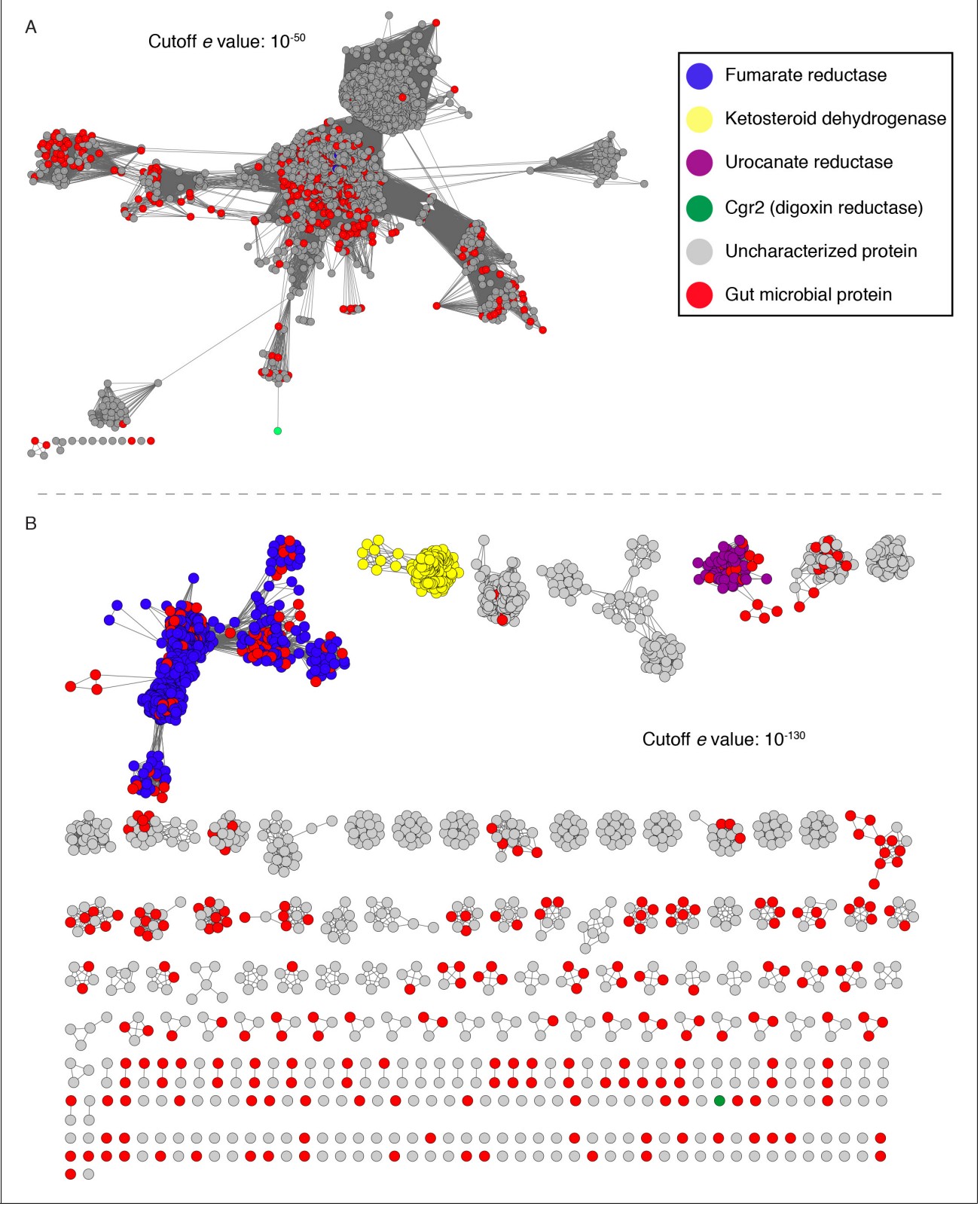

**Figure 5.** Sequence similarity network (SSN) analysis reveals that the gut bacterial enzyme Cgr2 is a highly distinct member of a large enzyme family that is widespread in gut microbes. The SSN was constructed using the top 5000 most similar proteins to Cgr2 from the UniprotKB database. Nodes represent proteins with 100% sequence identity. (**A**) SSN displayed with an e-value threshold of $10^{-50}$. The seven previously characterized enzymes (PDB ID: 1D4D, 1E39; UniProtKB ID: Q07WU7, Q9Z4P0, 8CVD0, P71864, Q7D5C1) and Cgr2 are colored according to biochemical function. (**B**) SSN

*Figure 5 continued on next page*

*Figure 5 continued*

displayed with an e-value threshold of $10^{-130}$. All nodes that co-clustered with characterized enzymes are shown in the same color, denoting putative isofunctional activity. With the exception of Cgr2, if a node comes from a gut bacterium, it is colored red rather than the color of the corresponding cluster.

DOI: https://doi.org/10.7554/eLife.33953.026

The following source data and figure supplements are available for figure 5:

**Source data 1.** Characterized enzymes within the Cgr2 sequence similarity network.

DOI: https://doi.org/10.7554/eLife.33953.031

**Figure supplement 1.** Multiple sequence alignment of fumarate reductases.

DOI: https://doi.org/10.7554/eLife.33953.027

**Figure supplement 2.** Multiple sequence alignment of urocanate reductases.

DOI: https://doi.org/10.7554/eLife.33953.028

**Figure supplement 3.** Multiple sequence alignment of ketosteroid dehydrogenases.

DOI: https://doi.org/10.7554/eLife.33953.029

**Figure supplement 4.** Cgr2 is a distinct flavoprotein reductase.

DOI: https://doi.org/10.7554/eLife.33953.030

these results suggest that Cgr2 activity is restricted to cardenolide toxins and does not extend to other structurally related endogenous or exogenous compounds.

Additionally, neither fumarate nor any of the metabolized cardenolides conferred a growth advantage to *cgr2+ E. lenta* in minimal or rich medias, suggesting that these compounds are not used as alternative terminal electron acceptors (*Figure 4—figure supplement 2*). The inability of Cgr2 to reduce fumarate, a common electron acceptor used during bacterial anaerobic respiration, led us to revisit the original annotation of Cgr2 as a 'fumarate reductase' (*Saunders et al., 2009*). To more systematically assess the relationship between Cgr2 and biochemically characterized reductases, we constructed a sequence similarity network (SSN) using the 5000 most similar sequences from the UniProtKB protein database. Within the network, there were seven enzymes that had been biochemically characterized (UniProtKB IDs: Q07WU7, Q9Z4P0, 8CVD0, P71864), biochemically and structurally characterized (PDB IDs: 1D4D, 1E39), or genetically characterized (UniProtKB ID: Q7D5C1) (*Figure 5—source data 1*) (*Leys et al., 1999*; *Bogachev et al., 2012*; *Brzostek et al., 2005*; *Doherty et al., 2000*; *Knol et al., 2008*; *Rothery et al., 2003*; *Pealing et al., 1992*; *Dobbin et al., 1999*). At all thresholds at which Cgr2 remained connected to other protein sequences, all characterized enzymes within the SSN were co-clustered, precluding the resolution of unique biochemical functions at this cutoff (*Figure 5A*). At higher alignment thresholds that separated these characterized enzymes into discrete isofunctional clusters, Cgr2 was always present as a 'singleton' with no links to other protein sequences (*Figure 5B*). Our SSN also revealed that reductase enzymes are widespread among human gut bacteria, with three validated enzymatic activities and 113 distinct clusters with uncharacterized biochemical functions detected in sequenced gut bacterial genomes.

We validated our SSN by aligning the biochemically characterized enzymes within the network with additional co-clustered sequences to assess conservation of essential active site residues (*Figure 5—figure supplements 1–3*). Comparing the sequence of Cgr2 with sequences of biochemically characterized reductases revealed that Cgr2 lacks active site residues required for the activity of fumarate reductases (6/7 divergent residues), urocanate reductases (4/5 divergent residues), and ketosteroid dehydrogenases (3/5 divergent residues) (*Figure 5—figure supplement 4A–D*) (*Leys et al., 1999*; *Rohman et al., 2013*; *Bogachev et al., 2012*; *Knol et al., 2008*; *Reid et al., 2000*). Individually mutating the two residues shared between Cgr2 and ketosteroid dehydrogenases confirmed that one amino acid involved in substrate binding (G536 backbone) was also important for the activity of Cgr2 but the other (Y532) was not (*Figure 5—figure supplement 4E*). Together, the location of Cgr2 within the SSN and the differences in its sequence indicate that this enzyme is distinct from characterized bacterial reductases and may use a unique set of residues to catalyze cardenolide reduction.

## Cgr2 is widespread in the human gut microbiome

To assess the broader relevance of this cardenolide-metabolizing enzyme, we quantified the prevalence, conservation, and genomic context of *cgr2* in the human gut microbiome. We mined gut

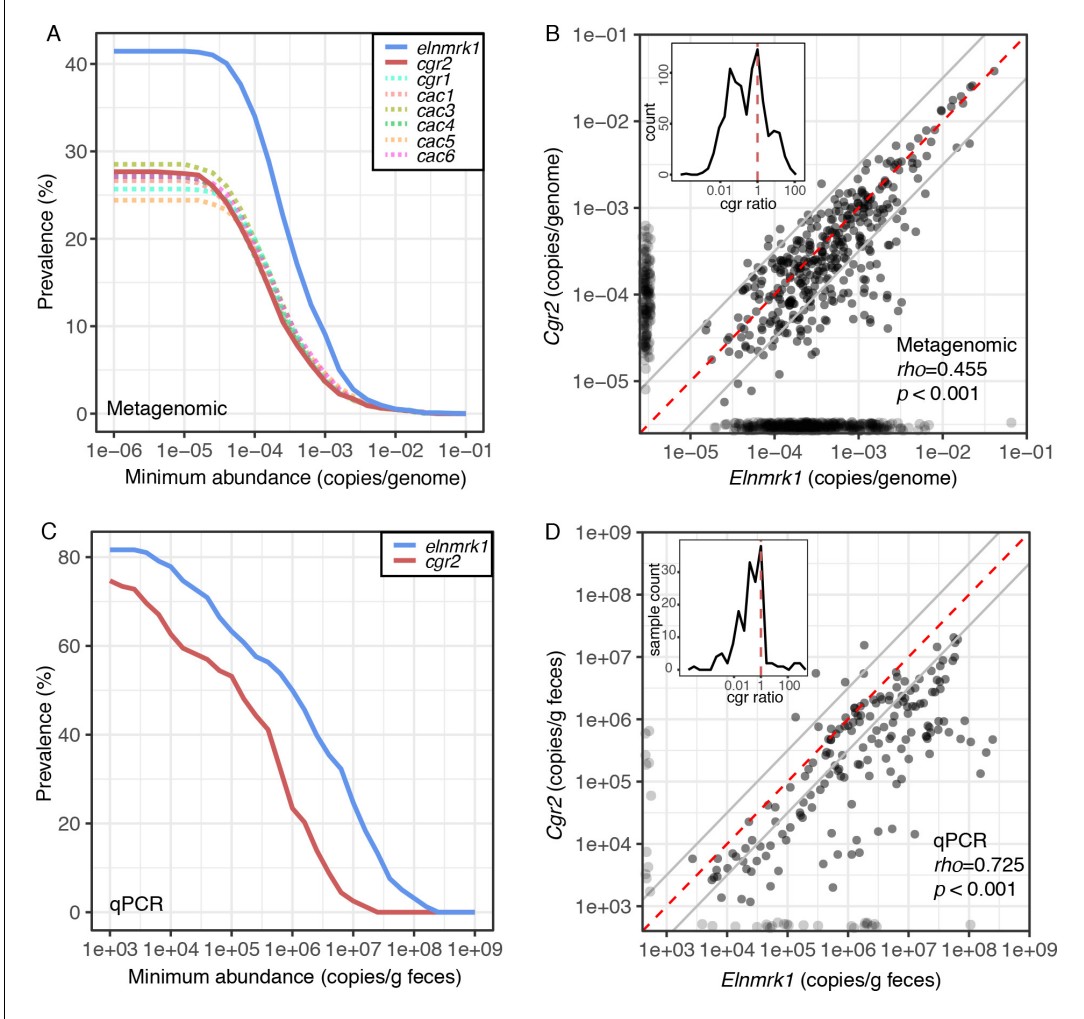

**Figure 6.** Cgr2 is widespread in the human gut microbiome. (A) Analysis of the *cgr*-associated gene cluster and *E. lenta* (via *elenmrk1*) prevalence in the gut metagenomes of 1872 individuals (*see* Materials and methods) revealed that both *E. lenta* and *cgr2* are highly prevalent (41.5% and 27.7% respectively) but frequently low in abundance. (B) Quantification of *E. lenta* and *cgr2* abundances in individual gut metagenomes revealed a tight correlation between the two, providing evidence that *cgr2* is restricted to *E. lenta* and that individuals may harbor sub-populations of both *cgr2+ and cgr2-* strains. Red line denotes the expected linear relationship and dashed lines represent a ± half log deviation. (Inset) Histogram of *cgr*-ratio (*cgr/ elnmrk1*) demonstrates a significant skew away from communities that would have more *cgr2* than expected by *E. lenta* abundance (p<0.001, D'Agostino skewness test). (C) Replication in an additional 158 individuals located in the USA (n = 85) and Germany (n = 73) via duplexed qPCR increased prevalence estimates to 74.7% and 81.6% at the extremes of detection limit (1e3 copies/g). qPCR samples were run in technical triplicate (*Figure 6—source datas 1* and *2*). (D) Similarly, qPCR-derived abundances of *E. lenta* and *cgr2* were correlated, corroborating metagenome-based analysis. (Inset) Histogram of *cgr*-ratio demonstrating significant skew (p<0.001).

DOI: https://doi.org/10.7554/eLife.33953.032

The following source data is available for figure 6:

**Source data 1.** qPCR based *E. lenta* and *cgr2* prevalence.
DOI: https://doi.org/10.7554/eLife.33953.033
**Source data 2.** Replicate qPCR data of for *cgr2* and *elnmrk1* in human fecal samples.
DOI: https://doi.org/10.7554/eLife.33953.034

microbiome datasets from 1872 individuals sampled in 6 countries across three continents (*Nayfach et al., 2015*). Analysis of the *E. lenta* pan-genome led to the discovery of a single copy marker gene (referred to here as *elnmrk1*) conserved in all sequenced strains that serves as a proxy for *E. lenta* abundance in both sequencing and quantitative PCR (qPCR) assays (*Bisanz et al., 2018*). The abundance of *elnmrk1* was significantly associated with *E. lenta* abundance across the

individuals ($R^2$ = 0.973, p<2.2e-16). Using this marker gene, we detected *E. lenta* in 41.5% of subjects at abundances of $-3.5 \pm 0.58$ mean ± SD $\log_{10}$(elnmrk1 copies/cell) (*Figure 6A*).

*Cgr2* was detectable in 48.5% of the *E. lenta*-positive individuals, while detection occurred in 27.7% of all subjects ($-3.7 \pm 0.64$ mean ± SD $\log_{10}$(cgr2 copies/cell)) (*Figure 6A*), and the abundance of *E. lenta* and *cgr2* was significantly associated (rho = 0.455, p<0.001; *Figure 6B*). The distributions were skewed towards participants with less *cgr2* than expected based on the abundance of *E. lenta* [(skew = $-0.773$, p=2.5e-8, n = 375, D'Agostino skewness test of log10(cgr/elnmrk1) with quantifiable *elnmrk1* and *cgr2*], consistent with prior data suggesting that many individuals are colonized by a mixture of *cgr2+* and *cgr2-* *E. lenta* strains (*Haiser et al., 2013*). These results were validated by qPCR in an independent set of 158 individuals (228 samples) from multiple sites in the USA and Germany (*Figure 6—source data 1*) revealing a similar skew towards higher *E. lenta* abundances versus *cgr2* (skew = $-0.65$, p<0.001, n = 165). Using this more sensitive detection method, we detected *E. lenta* in 81.6% of individuals (1.5e7 ± 3.5e6 copies/g feces) and *cgr2* in 74.7% (1.5e6 ± 3.5e6 copies/g feces) (*Figure 6C–D*). Similar to the sequence-based analysis, the outliers were skewed towards

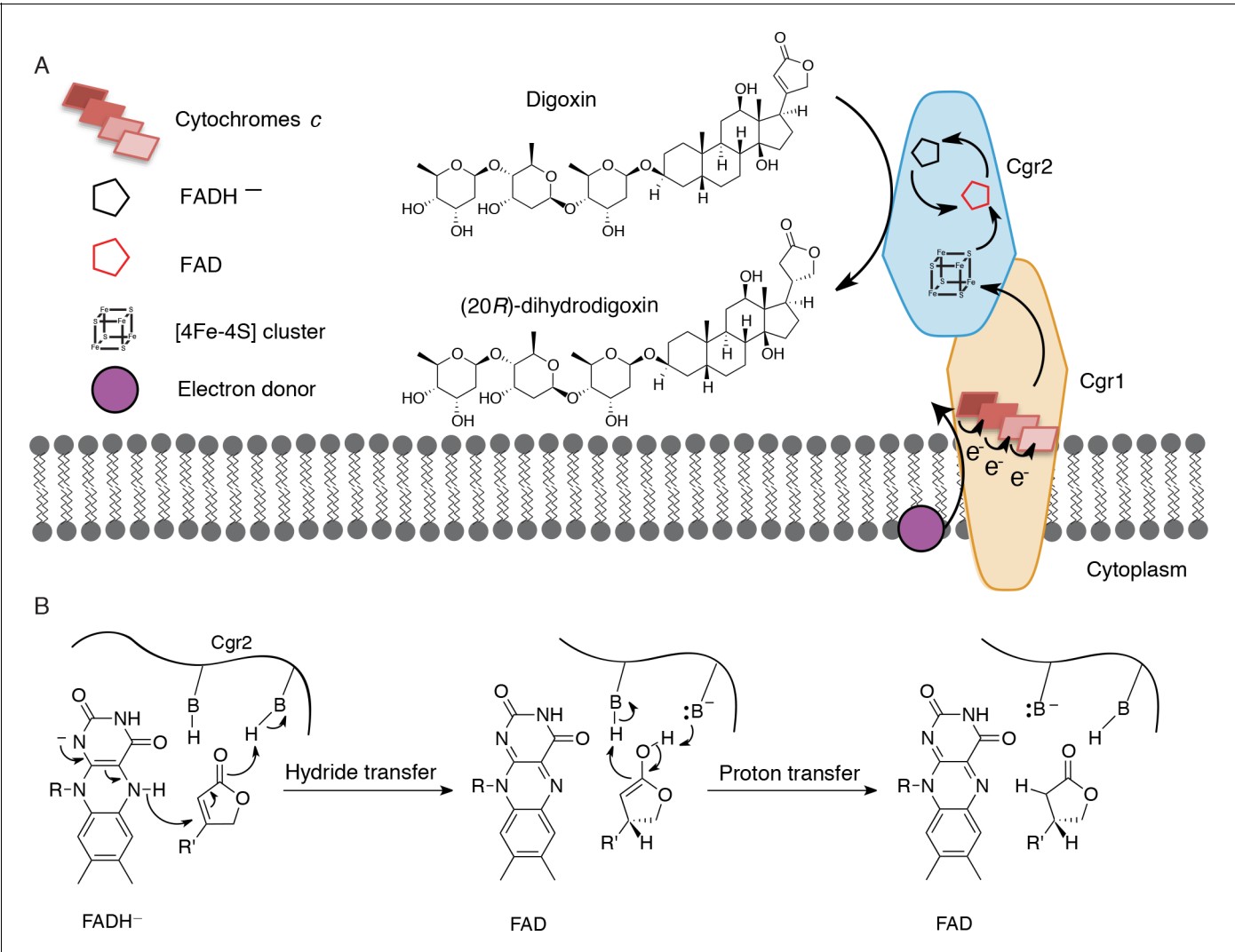

**Figure 7.** Preliminary model for digoxin metabolism by Cgr1 and Cgr2. (A) Proposed biochemical model and (B) mechanism of digoxin reduction by Cgr proteins. Cgr1 is predicted to transfer electrons from a membrane-associated electron donor to the $[4Fe-4S]^{2+}$ cluster of Cgr2 via covalently bound heme groups. The reduced $[4Fe-4S]^{1+}$ cluster of Cgr2 could sequentially transfer two electrons to FAD, generating FADH⁻, which could mediate hydride transfer to the β-position of the digoxin lactone ring. Protonation of the resulting intermediate would yield (20R)-dihydrodigoxin.
DOI: https://doi.org/10.7554/eLife.33953.035

samples with less *cgr2* than expected based on the abundance of *E. lenta*. Overall, both the qPCR- and metagenomic sequencing-based analyses show that *E. lenta* and *cgr2* are widely distributed in the human microbiome.

## Discussion

For over three decades, the human gut bacterium *E. lenta* has been linked to cardiac drug inactivation (*Saha et al., 1983*). However, the identity, specificity, and distribution of the enzymes responsible for this activity were unknown. In this work, we unambiguously show that the *E. lenta* protein, Cgr2, inactivates cardenolides, including the pharmaceutical agents digoxin and digitoxin that have been used for over two centuries in the treatment of cardiac diseases. Although we cannot rule out the possibility that other enzymes are involved in digoxin inactivation, no other microbes have been discovered that possess this metabolic activity apart from *cgr2+ E. lenta* (*Saha et al., 1983*). Cgr2 represents a novel flavoprotein reductase, and contains oxygen-sensitive [4Fe-4S] cluster(s) and a divergent set of predicted active site residues. The failure of bioinformatic analyses to identify this essential [4Fe-4S] cluster highlights the need for additional structural and mechanistic studies of the *cgr* operon and other gut microbial enzymes involved in xenobiotic metabolism. Our working model is that Cgr1 and Cgr2 form a membrane-anchored, extracellular complex that mediates electron transfer from an electron donor (e.g. the membrane quinone pool) through multiple cytochromes *c* in Cgr1 to Cgr2, which ultimately reduces the α,β-unsaturated γ-butyrolactone of digoxin and other cardenolides (*Figure 7A*). We propose that the $[4Fe-4S]^{2+}$ cluster(s) of Cgr2 sequentially transfer electrons to the FAD cofactor. The resulting hydride equivalent is then transferred to the cardenolide, and proton transfer generates the fully reduced γ-butyrolactone (*Figure 7B*), yielding the therapeutically inactive metabolite dihydrodigoxin.

While we have demonstrated that Cgr2 is necessary and sufficient for digoxin reduction in a heterologous host and in vitro using a chemical electron donor, additional proteins within the expanded *cgr* gene cluster may be important for digoxin reduction in vivo. Sequence analyses and transcriptional data suggest that Cgr1 is likely important for this metabolic activity in *E. lenta.* However, we were unable to observe overexpression or heme *c* incorporation into Cgr1 using a variety of heterologous constructs, hosts, and expression conditions, which may be due to an incompatibility of heterologous cytochrome *c* maturation factors and this protein (*Sanders and Lill, 2000*; *Thöny-Meyer, 2002*). The use of alternative heterologous systems that are more suitable for producing multi-heme cytochromes *c* (*Ozawa et al., 2001*; *Kern and Simon, 2011*) or the development of genetic tools in *E. lenta* would thus be required to obtain functional Cgr1 and determine its role in digoxin metabolism.

In contrast to Cgr1, the relevance of the Cac proteins to the Cgr proteins and digoxin metabolism is unclear. Apart from the putative LuxR-type regulator Cac3, which was modestly upregulated in response to digoxin and may be involved in regulating transcription of the *cgr* operon, RNA sequencing does not show clear evidence for the expression of the other *cac* genes during growth in pure culture. While Cac4 is annotated as a secreted, FAD-dependent fumarate reductase, it lacks all known catalytic and binding residues for this enzyme class (*Figure 5—figure supplement 1*). Furthermore, Cac4 shares only 23% sequence identity to Cgr2, and is thus likely to metabolize a different substrate than either of these enzymes. Cac6 is homologous to stomatin, prohibitin, flotillin, and HflK/C (SPFH) proteins, which are often associated with lipid rafts or functional microdomains in bacteria (*Bramkamp and Lopez, 2015*) and could potentially interact with the Cgr enzymes, the substrates of the *cgr* operon, or additional steroidal substrates of *E. lenta* (*Ridlon et al., 2006*; *Devlin and Fischbach, 2015*). Additional work is required to understand the biochemical function of the Cac proteins and whether they influence *E. lenta* metabolism of digoxin and/or other small molecules.

We have demonstrated that *E. lenta* strains harboring the *cgr* operon are widespread in the human gut microbiome, which supports the high incidence of dihydrodigoxin production observed clinically. Strikingly, *cgr2* and its associated genes are highly conserved, with two naturally occurring variants (frequent: Y333/N333; infrequent: V381/M381). This conservation is surprising given the strict specificity of Cgr2 towards cardenolides, which are ingested at very low concentrations to minimize toxicity in the context of cardiac therapy (*Smith et al., 1969*; *Gheorghiade et al., 2004*). These results, together with the overall high degree of conservation in the *E. lenta* pan-genome, suggest

that much of the phenotypic variation within this species may be driven by gene gain/loss rather than by genetic polymorphisms. Data from other bacterial lineages suggests that this phenomenon may not be unique to *Coriobacteriia* (*Hao and Golding, 2006*; *Marri et al., 2006*; *Marri et al., 2007*; *Nowell et al., 2014*; *Vos et al., 2015*); in *Pseudomonas syringae,* 1% amino acid divergence accumulates at the same time in hundreds or even thousands of genes (*Nowell et al., 2014*).

The high sequence conservation and levels of *cgr* operon transcription in response to digoxin exposure suggest that digoxin metabolism may provide a physiological benefit to *E. lenta*. However, as we could not observe any direct benefit of cardenolide metabolism for *cgr+ E. lenta* (*Figure 4— figure supplement 2*), we hypothesize that these bacteria may have evolved to protect the host from plant toxins and thus maintain a habitat for colonization. Although we cannot rule out an as-of-yet unidentified endogenous substrate of Cgr2, our results suggest that similar to intestinal and hepatic enzymes, gut bacteria have co-evolved with the human host and maintain detoxification systems that can be rapidly and efficiently mobilized on demand. Additional studies are warranted that directly compare and contrast the gut microbiomes of herbivorous animals and insects that evolved under more constant exposure to cardenolides (*Agrawal et al., 2012*), as these communities may represent a reservoir for evolutionary or functional homologs of *cgr2*. While we do not yet understand the factors that maintain this gene in the *E. lenta* population in the absence of a direct selective pressure, our studies, coupled with ex vivo experiments with human fecal samples and in vivo experiments in mice (*Haiser et al., 2013*), suggest that the metabolic activities of low abundance members of the gut microbiome can significantly influence host physiology.

Our results also highlight an important consideration for ongoing efforts to predict and manipulate gut microbial metabolism, particularly in the context of therapeutics. Not only do *E. lenta* strains vary in the presence or absence of *cgr2*, but we have also identified a naturally occurring, single amino acid substitution that causes a dramatic loss of activity in the Cgr2 enzyme. This result emphasizes the need for methods that can achieve nucleotide-level precision in mapping inter-individual differences in human gut microbiome gene content. Several clinical trials have recently investigated the use of digoxin for treating diverse cancers (*Lin et al., 2014*; *Kayali et al., 2011*), rheumatoid arthritis (*Huh et al., 2011*) and HIV-1 infection (*Wong et al., 2013*). Paired studies of host genetics combined with nucleotide-resolution analyses of the gut microbiome are needed to test the feasibility of using microbial genetic information to predict drug bioavailability and improve treatment outcomes in these new disease contexts.

Finally, our results demonstrate the feasibility of progressing from case studies of a clinically relevant gut microbial biotransformation (*Saha et al., 1983*) to identifying the responsible genes, enzymes, and biochemical mechanisms associated with metabolism. Isolation of individual xenobiotic metabolizing strains is a crucial first step towards uncovering the genetic and biochemical bases of these transformations. Starting from complex microbial communities (*e.g.* human fecal samples), individual strains can be selectively enriched (*Kumano et al., 2016*; *Martínez-del Campo et al., 2015*) or isolated (*Saha et al., 1983*; *Bisanz et al., 2018*) to enable screening and identification of microbes that metabolize a xenobiotic of interest. The observation that many xenobiotic-processing or transporting genes are only upregulated in the presence of substrate can be leveraged to identify xenobiotic-metabolizing genes using techniques such as RNA-seq (*Haiser et al., 2013*) or native protein purification (*Kumano et al., 2016*). Genetic and/or heterologous expression experiments can then help to validate the role of the identified genes and enzymes in xenobiotic transformation. Finally, the responsible genes can serve as candidate biomarkers to probe the distribution and potential for xenobiotic metabolism in ex vivo incubations or in relevant clinical populations.

These studies also provide new insights into the chemistry made possible by complex host-associated microbial communities. The gut microbiome encodes over 3 million genes (*Qin et al., 2010*) and >50% have unknown functions (*Human Microbiome Project Consortium, 2012*). Our study of just one highly unique, clinically relevant gut microbial enzyme has illuminated underappreciated functional diversity within the broader flavin-dependent reductases, which is widespread among human gut microbes (*Figure 5*). As reductive transformations represent a major route by which gut microbes metabolize xenobiotics, these unique, putative reductase enzymes provide a promising starting point for identifying additional gut microbiome-xenobiotic interactions. The approaches described here, together with high-throughput methods to characterize protein subfamilies and advances in bacterial culturing and genetic tools (*Goodman et al., 2011*; *Lim et al., 2017*), are

beginning to unlock the genetic 'dark matter' of the human gut microbiome as well as its critical role in the etiology and treatment of human disease.

# Materials and methods

## Key resources table

| Reagent type (species) or resource | Designation | Source or reference | Identifiers | Additional information |
|---|---|---|---|---|
| Gene (*Eggerthella lenta*) | *cgr1* | PMCID: PMC3035228 | *cgr1* | |
| Gene (*Eggerthella lenta*) | *cgr2* | PMCID: PMC3035228 | *cgr2* | |
| Gene (*Eggerthella lenta*) | 16S rRNA | Ref 11 | | |
| Gene (*Eggerthella lenta*) | *E. lenta* marker gene | Ref 11 | *elnmrk1* | |
| Strain, strain background (*Escherichia coli*) | One Shot Top10 | Thermo Fisher Scientific | | |
| Strain, strain background (*Rhodococcus erythropolis*) | L88 | doi: 10.1128/JB. 187.8.2582–2591.2005 | | |
| Strain, strain background (*Eggerthella lenta*) | *Eggerthella lenta* strains | Ref 11 | El1 - El21 | see *Figure 1—source data 1* for full descriptions |
| Strain, strain background (*Eggerthella sinensis*) | *Eggerthella sinensis* DSM16107 | Ref 11 | Es1 | |
| Strain, strain background (*Gordonibacter*) | *Gordonibacter* strains | Ref 11 | Gs1, Gs2 | |
| Strain, strain background (*Paraeggerthella hongkongensis*) | *Paraeggerthella hongkongensis* | Ref 11 | Ph1 | |
| Recombinant DNA reagent | pTip expression vectors | doi: 10.1128/AEM .70.9.5557–5568.2004 | | |
| Sequence-based reagent | Amplification *cgr1, cgr2* | Integrated DNA technologies | | see Table 1 for primers and constructs |
| Sequence-based reagent | Point mutants of Cgr2 | Integrated DNA technologies | | see Table 2 for primers and mutants |
| Sequence-based reagent | *cgr2* sequencing primer | This work | | Confirmed sequence of *E. lenta* isolates with primers: cgr2_fwd (TGCAATCAAGACAACCACGA), cgr2_internal (TCGGTGTACAACCACAATGC), and cgr2_rev (GTTGCGCTGTGATTAGACTG) |
| Sequence-based reagent | qPCR primers *cgr2* | This work | | cgr2_F (GAGGCCGTCGATTGGATGAT), cgr2_R (ACCGTAGGCATTGTGGTTGT), and cgr2_probe ([HEX]CGACACGG AGGCCGATGTCG[BHQ1]) |
| Sequence-based reagent | qPCR primers *elnmrk* | This work | | ElentaUni_F (GTACAACATGCTCCTTGCGG), ElentaUni_R (CGAACAGAGGATCGGGATGG), ElentaUni_Probe ([6FAM]TTCTGGCTGC ACCGTTCGCGGTCCA[BHQ1]), |
| Chemical compound, drug | BBL Brain Heart Infusion (BHI) media | Becton Dickinson | BD:L007440 | |
| Chemical compound, drug | L-arginine | Sigma Aldrich | SA:A5006 | |

*Continued on next page*

*Continued*

| Reagent type (species) or resource | Designation | Source or reference | Identifiers | Additional information |
|---|---|---|---|---|
| Chemical compound, drug | Digoxin | Sigma Aldrich | SA:D6003 | |
| Chemical compound, drug | Dihydrodigoxin | doi: 10.1126/science.1235872 | | |
| Chemical compound, drug | Digitoxin | Sigma Aldrich | SA:D5878 | |
| Chemical compound, drug | Digoxigenin | Sigma Aldrich | SA:D9026 | |
| Chemical compound, drug | Ouabain | Sigma Aldrich | SA:O3125 | |
| Chemical compound, drug | Ouabagenin | Sigma Aldrich | SA:O2627 | |
| Chemical compound, drug | Sypro Orange protein gel stain | Thermo Fisher Scientific | SA:S6650 | |
| Chemical compound, drug | Thiostrepton | Sigma Aldrich | SA:T8902 | |
| Chemical compound, drug | Methyl viologen | Sigma Aldrich | SA:856177 | |
| Chemical compound, drug | Sodium dithionite | Sigma Aldrich | SA:157953 | |
| Chemical compound, drug | FAD | Sigma Aldrich | SA:F6625 | |
| Chemical compound, drug | Iron (II) ammonium sulfate hexahydrate | Sigma Aldrich | SA:F1543 | |
| Chemical compound, drug | Sodium sulfide nonahydrate | Sigma Aldrich | SA:208043 | |
| Chemical compound, drug | Dithothreitol | Sigma Aldrich | SA:D0632 | |
| Chemical compound, drug | HEPES | Sigma Aldrich | SA:RDD002 | |
| Software, algorithm | Cytoscape | doi:10.1101/gr.1239303 | | |
| Software, algorithm | Prism software | Prism software | Graphpad Software v 7 | |
| Other | Anaerobic chambers | Coy Laboratory products; Mbraun | | |
| Other | LC-MS/MS | Agilent | Agilent:6410 Triple Quad LC/MS | |
| Other | Electron paramagentic resonance (EPR) spectrometer | Bruker | | |
| Other | CFX96 Touch Real-Time PCR machine | Bio-Rad | | |
| Other | PowerWave HT Microplate Spectrophotometer | BioTek | | |

## Genome analysis

Publically available genomes were retrieved from NCBI (*E. lenta* DSM 2243, PRJNA21093; *E. lenta* FAA1-3-56 PRJNA40023). New isolates were sequenced as described elsewhere (Bisanz, et al. 2018). Genomes were assembled with SPAdes 3.11.1 (*84*) and annotated with Prokka 1.12 (*Seemann, 2014*). All *E. lenta* strains studied were identified as *E. lenta* based on 16S rRNA sequencing and were de-replicated at the strain level by considering a pairwise average nucleotide identity (ANI) >99.99% as the same strain (github.com/widdowquinn/pyani). The maximum and minimum

ANI between studied *E. lenta* strains were 97.9% and 99.6% respectively. The phylogenetic tree was prepared using a set of 400 conserved proteins (*Segata et al., 2013*) rooted on the *Gordonibacter* strains. Newly sequenced strains were included as part of Bioproject PRJNA412637.

Global nucleotide and amino acid identity and related statistics were determined via the Needleman-Wunsh implementation in the pairwiseAlignment function of Biostrings with percentage identity calculated as 100*identical positions/(aligned positions + internal gaps). Statistical and graphical analysis was carried out using R 3.4.0. RNA sequencing reanalysis was carried out by mapping reads to the reference genome with Bowtie2 (*Langmead and Salzberg, 2012*), counting with HTSeq (*Anders et al., 2015*), and differential expression analysis using DESeq2 (*Love et al., 2014*). Original sequence data is available from the SRA with project identifier SRP018311.

For the purposes of comparative genomics, gene conservation was calculated by first clustering into orthologous clusters with proteinortho5 (*Lechner et al., 2011*) with a minimum 60% amino acid identity and 80% coverage. A presence/absence matrix de-replicated for co-occurring features was then used as the input for a random forest classifier (randomForest 4.6–12). Variable importance (mean decrease GINI) was used to extract the 15 most important features. A tool for this comparative genomic analysis is available as ElenMatchR (jbisanz.shinyapps.io/elenmatchr; copy archived at https://github.com/elifesciences-publications/ElenMatchR) (*Bisanz and Turnbaugh, 2018*) with digoxin reduction available as a demonstration dataset.

## Bacterial culturing

*Eggerthella lenta* and related strains were grown in BBL Brain Heart Infusion (BHI) media (BD, Franklin Lakes, NJ) supplemented with L-arginine (Sigma-Aldrich, St. Louis, MO) under an atmosphere of 2–5% $H_2$, 2–5% $CO_2$, and balance $N_2$. Strains were streaked onto BHI agar plates supplemented with 1% arginine (w/v) in an anaerobic chamber (Coy Laboratory Products, Grass Lakes, MI). Individual colonies were inoculated into $16 \times 125$ mm Hungate tubes (Chemglass Life Sciences, Vineland, NJ) containing 5–10 mL of BHI supplemented with 1% arginine and grown at 37°C for 2–3 days. Cardiac glycoside substrates were dissolved at a concentration of 10 mM in dimethylformamide (DMF) and added to cultures at a final concentration of 10 µM. Starter cultures were diluted into 10 mL of BHI + substrate to a starting of $OD_{600}$ of 0.05 and grown anaerobically at 37°C for 2 days. Experiments were performed in biological triplicate.

For growth assays, *E. lenta* DSM 2243 was grown in either rich (BHI) or defined media. Basal media lacking terminal electron acceptors was prepared as previously described (*Löffler et al., 2005*) with the following modifications: yeast extract and tryptone were each added to 0.1% (w/v), L-cysteine concentration was 0.4 mM, sodium sulfide was not added, and either 5% $H_2$ or 10 mM sodium acetate were used as electron donors. Starter cultures were prepared as described above in BHI media supplemented with 1% arginine, and diluted 1:100 into media that had been supplemented with substrates (dissolved in DMF) to a final concentration of 10 µM. Cultures were grown anaerobically at 37°C in biological triplicate. $OD_{600}$ measurements were recorded on a Genesys20 spectrophotometer (Thermo Fisher Scientific, Waltham, MA).

## Extraction and LC-MS/MS detection of digoxin and dihydrodigoxin

Bacterial cultures were centrifuged (10 min x 4000 rpm), 1 mL of supernatant was extracted three times with 1 mL of dichloromethane and the pooled organic fractions were concentrated using a rotary evaporator. Samples were resuspended in 1 mL of 50% methanol in water and diluted 10x prior to liquid chromatography-tandem mass spectrometry (LC-MS/MS) analysis.

Metabolites were detected on an Agilent 6410 Triple Quad LC/MS using electrospray ionization in negative ion mode. The mass spectrometer settings were as follows: gas temperature (300°C), gas flow (10 L/min), nebulizer pressure (25 psi), capillary voltage (4000 V), and chamber current (0.1 µA). Digoxin was monitored using a 779.4 → 649.3 m/z transition with a fragmentor voltage of 250V and collision energy of 52, and dihydrodigoxin was monitored using a 781.4 → 521.3 m/z transition with a fragmentor voltage of 200V and collision energy of 20. Standard curves were prepared using 0.01–1 µM samples of each compound. Digoxin was purchased from Sigma-Aldrich (St. Louis, MO), and a dihydrodigoxin standard was obtained through chemical hydrogenation of digoxin as previously described (*Haiser et al., 2013*). Liquid chromatography was performed on an Acclaim Polar Advantage II column with a flow rate of 0.125 mL/min, 5 µL sample injection, solvent A (10% methanol + 1

**Table 1.** Primers and constructs for heterologous expression of Cgr proteins in *R. erythropolis*. Restriction sites are bolded.

| Construct | For/Rev | Sequence | Restriction sites | Vector | Anneal temp (°C) | Extend time (s) |
|---|---|---|---|---|---|---|
| *cgr* operon | For | ACTGACCCATGGATGGAATACGGAAAGTGCC | n/a | n/a | 71 | 75 |
| | Rev | GTTTTACTGCAGTTACGCCGCCGTCGAA | | | | |
| Cgr1 + Cgr2 | For | TGAC**GAATTC**TAATGGAATACGGAAAGTGCCG | EcoRI, | pTipQT2 | 70 | 90 |
| | Rev | TTATA**AGATCT**CGCCGCCGTCGAAAG | BglII | | | |
| Cgr1 | For | TCGAA**CATATG**GATGGCTGAGGAACCTGTGG | NdeI, | pTipQT1 | 65 | 60 |
| | Rev | ATAA**CTCGAG**TCACGCCGCCGTCGAAA | XhoI | | | |
| Cgr2 (native) | For | ACTGAC**CCATGG**GCATGGAATACGGAAAGTGCC | NcoI, HindIII | pTipQC2 | 65 | 60 |
| | Rev | ATTAG**AAGCTT**TCACTCCCACGGCTCGAG | | | | |
| Cgr2-CHis$_6$ (native) | For | ACTGAC**CCATGG**GCATGGAATACGGAAAGTGCC | NcoI, HindIII | pTipQC1 | 65 | 60 |
| | Rev | GTTAG**AAGCTT**CTCCCACGGCTCGAG | | | | |
| Cgr2 (−48aa)-NHis$_6$ | For | TATTA**CCATGG**ATCAGACCGCGCCTGC | NcoI, HindIII | pTipQC2 | 65 | 60 |
| | Rev | ATACT**AAGCTT**CTCCCACGGCTCGA | | | | |
| Cgr2 (−48aa)-CHis$_6$ | For | TATTA**CCATGG**ATCAGACCGCGCCTGC | NcoI, HindIII | pTipQC1 | 65 | 60 |
| | Rev | ATACT**AAGCTT**TTACTCCCACGGCTCGA | | | | |
| Sequencing primers | For | CGTGGCACGCGGAAC | n/a | All pTip vectors | n/a | n/a |
| | Rev | GTGCAGGTTTCGCGTG | | | | |

DOI: https://doi.org/10.7554/eLife.33953.036

mM ammonium hydroxide) and solvent B (100% methanol + 1 mM ammonium hydroxide), and a gradient: 70–100% B over 10 min, 100% B for 1.5 min, 100–70% B over 3.5 min, and 70% B for 7 min.

## Construction of *cgr1* and *cgr2* vectors in *Escherichia coli*

*E. lenta* DSM 2243 was grown in 5 mL of BHI + 1% arginine at 37°C. After 2 days, the culture was pelleted and genomic DNA (gDNA) was purified with the UltraClean Microbial DNA Isolation Kit (QIAGEN, Germantown, MD) according to the manufacturer's protocol. The *cgr* operon was amplified from 50 ng of gDNA in a 50 µL reaction volume with 0.5 µM of each primer (*Table 1*) and Phusion High-Fidelity PCR master mix with HF buffer (New England Biolabs, Ipswich, MA). The following thermocycling parameters were used: denaturation at 98°C for 3 min; 35 cycles of 98°C for 15 s, 71°C for 20 s, and 72°C for 75 s; and a final extension at 72°C for 5 min. The PCR reaction was analyzed by agarose gel electrophoresis (1% agarose gel), and the *cgr* amplicon was excised and purified with the Illustra GFX PCR DNA and Gel Band Purification kit (GE Healthcare, Chicago, IL). *Cgr1* and *cgr2* variants were amplified in 20 µL PCR reactions using 1 ng of purified *cgr* operon as template, 0.5 µM primer pairs and Phusion High-Fidelity PCR master mix with HF buffer (New England Biolabs, Ipswich, MA) (*Table 1*). PCR conditions were as follows: denaturation at 98°C for 2 min; 35 cycles of 10 s at 98°C, 10 s at specified annealing temperature, and 72°C for the specified extension time; and a final extension at 72°C for 5 min. *Cgr* amplicons were digested in a 30 µL reaction with 1.5 µL of each restriction enzyme (New England Biolabs, Ipswich, MA) for 2.5 hr at 37°C. pTip vectors were similarly digested, and the linearized vector was excised from a 1% agarose gel and purified. Insert and vector pairs were ligated at a 1:3 ratio at room temperature for 2 hr with T4 DNA ligase (New England Biolabs, Ipswich, MA). 2.5 µL of the ligation reaction was transformed into chemically competent One Shot Top10 *E. coli* cells (Thermo Fisher Scientific, Waltham, MA) and plated on LB agar with ampicillin. Plasmid inserts were sequenced using the primers listed in *Table 1*.

**Table 2.** Primers for site-directed mutagenesis of Cgr2.
Amino acid numbering is based on full length Cgr2 sequence. Introduced mutations are bolded.

| Mutant | F/R | Sequence |
|---|---|---|
| C82A | For | CAGCGGCGGCACG**GCC**GCGGCCATCG |
|  | Rev | CCTCGATGGCCGC**GGC**CGTGCCGCCG |
| C111A | For | GCGGCAACTCGGCACTA**GCC**GGTGGATACAT |
|  | Rev | CCAGCATGTATCCACC**GGC**TAGTGCCGAGTTG |
| C158A | For | ATATGATCCGCGAGGCG**GCC**TTGCGCTCCGGC |
|  | Rev | GCCTCGCCGGAGCGCAA**GGC**CGCCTCGCGGAT |
| C187A | For | GCCCCCGGTCTGGTCA**GCC**GGCGACACGG |
|  | Rev | GGCCTCCGTGTCGCC**GGC**TGACCAGACCGG |
| C231A | For | CGAAATCGAGATGGGC**GCC**GAGGTGGCGCAC |
|  | Rev | GATGTGCGCCACCTC**GGC**GCCCATCTCGAT |
| C265A | For | GGCGTGGTCATGGCG**GCC**GCTTCGGTGGA |
|  | Rev | GTTGTCCACCGAAGC**GGC**CGCCATGACCA |
| C321A | For | GATCGGTGCTGAGCTT**GCC**ATGCAGCAGGC |
|  | Rev | CACGGCCTGCTGCAT**GGC**AAGCTCAGCACC |
| C327A | For | CATGCAGCAGGCCGTG**GCC**ATGAACGATTCT |
|  | Rev | GATAGAATCGTTCAT**GGC**CACGGCCTGCTG |
| C371A | For | GACCGGCAGACGGTTT**GCC**CAGGACGATGCCG |
|  | Rev | CTCGGCATCGTCCTG**GGC**AAACCGTCTGCC |
| C384A | For | CTATGTCATGCACGAG**GCC**GCGCAAGCTGCA |
|  | Rev | CCATGCAGCTTGCGC**GGC**CTCGTGCATGAC |
| C425A | For | CATACGCCCGACACG**GCC**GATACTACGTTC |
|  | Rev | CGAGAACGTAGTATC**GGC**CGTGTCGGGCGT |
| C443A | For | GCCGAGTTTATCGGC**GCC**GATCCGACCGC |
|  | Rev | GAGGGCGGTCGGATC**GGC**GCCGATAAACTC |
| C459A | For | GAGGTGGAACTCTTTC**GCC**GAGGCCGGTTTG |
|  | Rev | CATCCAAACCGGCCTC**GGC**GAAAGAGTTCCA |
| C483A | For | GACGCCGCCGTTCTAC**GCC**GATGTCGTGCGC |
|  | Rev | GGGGCGCACGACATC**GGC**GTAGAACGGCGG |
| C521A | For | CTGTACGGCGCCGGG**GCC**ATCATCGGGGGT |
|  | Rev | GTTACCCCCGATGAT**GGC**CCCGGCGCCGTA |
| C535A | For | GCCTTCTACTTCGGC**GCC**GGCTGGTCCATC |
|  | Rev | CGTGATGGACCAGCC**GGC**GCCGAAGTAGAA |
| Y333N | For | GCATGAACGATTCTATC**AAC**GTAGGCGGCATCA |
|  | Rev | TCGCTGATGCCGCCTAC**GTT**GATAGAATCGTTCA |
| Y532F | For | GATGCCGAGTGGGGC**TTT**GTCATGCACG |
|  | Rev | GCACTCGTGCATGAC**AAA**GCCCCACTCG |
| G536A | For | TTCTACTTCGGCTGC**GCC**TGGTCCATCA |
|  | Rev | GTTCGTGATGGACCA**GGC**GCAGCCGAAG |

DOI: https://doi.org/10.7554/eLife.33953.037

## Site-directed mutagenesis of Cgr2

Site-directed mutagenesis was performed in 25 µL reactions using 200 ng of template DNA (Cgr2(–48aa)-NHis$_6$ in pTipQC2), 0.5 µM of each primer pair (*Table 2*), 0.5 mM dNTP, and 1 µL of Pfu Turbo

polymerase AD (VWR, Radnor, PA). The following thermocycling parameters were used: denaturation at 95°C for 1 min; 18 cycles of 95°C for 30 s, 65°C for 50 s, and 68°C for 22 min (2 min/kb); and a final extension at 68°C for 7 min. The template plasmid was digested with 1 μL of DpnI (New England Biolabs, Ipswich, MA) for 1 hr at 37°C, and 2 μL of the reaction were transformed into chemically competent One Shot Top10 *E. coli* cells (Thermo Fisher Scientific, Waltham, MA).

## Heterologous expression of Cgr proteins in *Rhodococcus erythropolis* L-88

All *Rhodococcus* strains and expression vectors were obtained from the National Institute of Advanced Industrial Science and Technology (Tokyo, Japan). 40 ng of plasmid DNA were added to 400 μL of *R. erythropolis* L-88 electrocompetent cells in 30% PEG 1000 (Sigma-Aldrich, St. Louis, MO) in a 2 mm gap electroporation cuvette (VWR, Radnor, PA). Cells were transformed in a Micro-Pulser electroporator (Bio-Rad, Hercules, CA) with a 2.5 kV pulse (time constant ~4.8 – 5.2), rescued with 0.6 mL of LB (Lennox) broth (Alfa Aesar, Tewksbury, MA), and incubated for 4 hr at 28°C, 175 rpm. Cells were plated onto LB agar plates +antibiotic (17 μg/mL chloramphenicol for pTipQC plasmids; 8 μg/mL tetracycline for pTipQT plasmids) and incubated at 28°C for 5–7 days. Single colonies were inoculated into 50–75 mL of LB +antibiotic (34 μg/mL chloramphenicol or 8 μg/mL tetracycline) and grown for 3–5 days at 28°C, 175 rpm until reaching saturation. For gain of function studies, 50 mL of LB and antibiotic were inoculated to a starting $OD_{600}$ of 0.2 and grown at 28°C, 175 rpm. Experiments were performed in biological triplicate. When cultures reached an $OD_{600}$ of 0.6 (~6–8 hr), protein expression was induced with thiostrepton (Sigma-Aldrich, St. Louis, MO) at a final concentration of 0.01 μg/mL, and cultures were incubated at 15°C, 175 rpm. In cultures where Cgr1 was overexpressed, media was supplemented with the heme precursor δ-amino levulinic acid hydrochloride (50 μg/mL final concentration) (Frontier Scientific, Logan, Utah). After 16–20 hr, digoxin was added to cultures as a solution in DMF at a final concentration of 10 μM and incubated for either 7 days at 15°C, or 2 days at 28°C, 175 rpm. Culture supernatants were extracted and analyzed as previously described. For large-scale purifications, 2 L of LB-chloramphenicol in a 4 L baffled flask were inoculated to a starting $OD_{600}$ of 0.02 and grown to an $OD_{600}$ of 0.6 (~18–25 hr). Protein expression was induced with 0.01 μg/mL thiostrepton, and cultures were incubated at 15°C, 175 rpm for approximately 21 hr before harvesting cells by centrifugation (10,800 rpm x 20 min). Cell pellets were frozen and stored at –80°C.

## Cgr2 purification and [Fe-S] cluster reconstitution

All protein purification steps were carried out at 4°C. Harvested cells were resuspended in 5 mL/g of cell pellet in lysis buffer (50 mM Tris, pH 8, 1 mM $MgCl_2$, 25 mM imidazole) containing Pierce EDTA-free protease inhibitor cocktail (Thermo Fisher Scientific, Waltham, MA). Cells were passaged through a cell disruptor (Avestin EmulsiFlex-C3) five times at 15,000–25,000 psi and centrifuged for 20 min at 13,000 rpm. The clarified lysate was incubated on a nutating mixer with 5–10 mL of HisPur Ni-NTA resin (Thermo Fisher Scientific, Waltham, MA) for 1 hr and then applied to a gravity flow column. The resin was washed with 50 mL of wash buffer (25 mM HEPES, 0.5 M NaCl, pH 8, 25 mM imidazole) and eluted with 25 mL of elution buffer (25 mM HEPES, 0.5 M NaCl, pH 8, 200 mM imidazole). Eluted protein was concentrated using a 20 mL Spin-X UF 30 k MWCO PES spin filter (Corning, Corning, NY) to a volume of 1–2.5 mL, and then desalted on a Sephadex G-25 PD-10 desalting column (GE Healthcare, Chicago, IL) that had been equilibrated with desalting buffer (50 mM HEPES, 100 mM NaCl, pH 8). Desalted protein was sparged with argon on ice for 30–45 min. Chemical reconstitution of [Fe-S] cluster(s) in Cgr2 was carried out at 4°C in an anaerobic chamber (Coy Laboratory Products, Grass Lakes, MI) under an atmosphere of 2% hydrogen and 98% nitrogen. A 30 μM solution of Cgr2 was prepared in reconstitution buffer (50 mM HEPES, 100 mM NaCl, pH 8, and 2 mM dithiothreitol (DTT)). $Fe(NH_4)_2(SO_4)_2 \cdot 6H_2O$ (Sigma-Aldrich, St. Louis, MO) was added in four aliquots over 60 min, followed by addition of $Na_2S \cdot 9H_2O$ (Sigma-Aldrich, St. Louis, MO) in four aliquots over 60 min to final concentrations of 0.24 or 0.375 mM (8 or 12.5 equivalents relative to Cgr2), and stirred for 16–24 hr. The reaction was filtered through a 0.25 mm, 0.2 μM pore-size PES syringe filter (VWR, Radnor, PA) to remove precipitant and concentrated in a 6 mL Spin-X UF 30 k MWCO PES spin filter inside a 50 mL conical-bottom centrifuge tube with plug seal cap (Corning, Corning, NY). The concentrated protein (1–2.5 mL) was desalted on a PD-10 column into desalting

buffer. Protein was aliquoted into 0.5 mL PP conical tubes with skirt (Bio Plas), sealed inside 18 × 150 mm Hungate tubes (Chemglass Life Sciences, Vineland, NJ) and stored at –80°C. Protein concentration was determined by Bradford using Protein Assay Dye Reagent (Bio-Rad, Hercules, CA) and bovine serum albumin (BSA) (Sigma-Aldrich, St. Louis, MO) as a reference standard. Typical protein yields were ~20 mg/L of culture for both wild-type and point mutants of Cgr2(–48aa)-NHis$_6$,~8 mg/L for Cgr2(–48aa)-CHis$_6$, and ~1 mg/L for Cgr2-CHis$_6$. The iron and sulfur content of Cgr2 samples (protein concentrations between 20–50 μM) was determined using previously reported colorimetric assays (*Craciun et al., 2014*).

## Thermal denaturation assays

Thermal denaturation assays of purified and reconstituted Cgr2 were prepared on ice in 0.2 mL skirted 96-well PCR plates (VWR, Radnor, PA) sealed with optical adhesive covers (Life Technologies, Woburn, MA). Each reaction contained 10 μg of purified or reconstituted Cgr2, Sypro Orange protein gel stain (Thermo Fisher Scientific, Waltham, MA) diluted 5000-fold, and buffer containing 100 mM buffering agent and 100 mM NaCl in a total volume of 30 μL. The following buffering agents were used: acetate/acetic acid for pH 4–6, HEPES for pH 7, Tris-HCl for pH 8–9, and glycine-NaOH for pH 10. For metal binding assays, metal salts (Sigma-Aldrich, St. Louis, MO) were dissolved in pH 8 buffer to generate 100 mM stock solutions and added to a final concentration of 48 μM (8 equivalents relative to Cgr2). Data was collected on a CFX96 Touch Real-Time PCR machine (Bio-Rad, Hercules, CA) using the 'FRET' filter setting with FAM excitation and HEX emission channels (485 nm and 556 nm respectively). The following temperature-scan protocol was used: 25°C for 30 s, then ramp from 25°C to 100°C at a rate of 0.1 °C/ s.

## Gel filtration

Gel filtration experiments were carried out on a Superdex 200 10/300 GL column (GE Healthcare, Chicago, IL) attached to a BioLogic DuoFlow chromatography system (Bio-Rad, Hercules, CA). Experiments were carried out either aerobically or anaerobically inside a Coy anaerobic chamber (Coy Laboratory Products, Grass Lakes, MI). 100 μL protein samples (50–100 μM) were loaded onto the column at a rate of 0.2 mL/min for 1 mL followed by an isocratic flow of 0.33 mL/min for 30 mL with 50 mM HEPES, 100 mM NaCl, pH 8. The molecular weight for Cgr2(–48aa)-NHis$_6$ is 55.7 kDa. A gel filtration standard (Bio-Rad, Hercules, CA) containing thyroglobulin (670 kDa), γ-globulin (158 kDa), ovalbumin (44 kDa), myoglobin (17 kDa), and vitamin B12 (1.35 kDa) was used to determine the molecular weight of Cgr2-containing peaks.

## UV-vis spectroscopy

Cgr2 was diluted to a final concentration of 50–100 μM in UV-Star UV-transparent 96-well microplates (Greiner Bio-One, Monroe, NC). The absorbance was measured between 250–750 nm using a PowerWave HT Microplate Spectrophotometer (BioTek, Winooski, VT) inside of an anaerobic glovebox (Mbraun, Stratham, NH). Curves were baseline subtracted using respective absorbance values at 700 nm. To determine whether the [Fe-S] cluster(s) were redox active, Cgr2 was incubated with 10 equivalents of sodium dithionite (Sigma-Aldrich, St. Louis, MO) for 15 min at room temperature prior to taking additional absorption spectra. To assess the oxygen sensitivity of [Fe-S] cluster(s), Cgr2 was taken out of the anaerobic chamber and exposed to oxygen, and the absorption spectra was measured aerobically on a PowerWave HT Microplate Spectrophotometer (BioTek, Winooski, VT). Oxygen-exposed Cgr2 was then sparged for 30 min with argon (on ice) and brought back into the Mbraun glovebox for activity assays.

## EPR spectroscopy

All samples were prepared in 50 mM HEPES, 100 mM NaCl, pH 8 under oxygen-free conditions in an anaerobic glovebox (Mbraun, Stratham, NH). For all EPR experiments the final concentration of Cgr2 was either 150 or 200 μM. When required, the samples were reacted with an excess of sodium dithionite (10–20 equivalents) for 20–30 min at 22°C prior to freezing in liquid N$_2$. Spin quantification was carried out against a Cu$^{2+}$-EDTA standard containing an equimolar concentration of CuSO$_4$ in 10 mM EDTA (150 or 200 μM), under non-saturating conditions. Samples (250 μL) were loaded into 250 mm length, 4 mm medium wall diameter Suprasil EPR tubes (Wilmad LabGlass, Vineland, NJ)

and frozen in liquid $N_2$ under oxygen-free conditions. EPR spectra were acquired on a Bruker E500 Elexsys continuous wave (CW) X-Band spectrometer (operating at approx. 9.38 GHz) equipped with a rectangular resonator (TE102) and a continuous-flow cryostat (Oxford 910) with a temperature controller (Oxford ITC 503). The spectra were recorded at variable temperatures between 10–40 K at a microwave power of 0.2 mW, using a modulation amplitude of 0.6 mT, a microwave frequency of 9.38 GHz, a conversion time of 82.07 ms, and a time constant of 81.92 ms.

## In vitro substrate reduction assays

Methyl viologen (paraquat) dichloride hydrate (Sigma-Aldrich, St. Louis, MO) that had been reduced with sodium dithionite was used as an artificial electron donor (*Watanabe and Honda, 1982*) to initiate anaerobic Cgr2-mediated reduction of digoxin in vitro. Assays were carried out at 25°C in an anaerobic glovebox (Mbraun, Stratham, NH) under an atmosphere of nitrogen and < 5 ppm oxygen. Reagents were brought into the glovebox as solids or sparged liquids and resuspended in anoxic buffer inside the chamber: flavin (FAD or FMN) and methyl viologen (MV) were resuspended in 50 mM HEPES, 100 mM NaCl, pH 7 to generate stock solutions of 1 mM and 50 mM respectively; sodium dithionite was resuspended in 50 mM HEPES, 100 mM NaCl, pH 8 to generate a stock solution of 25 mM; all substrates (*Figure 4—figure supplement 1*) were dissolved in DMF to generate stock solutions of 25 mM, with the exception of sodium fumarate dibasic and urocanic acid which were dissolved in water. All substrates and reagents were purchased from Sigma-Aldrich (St. Louis, MO) except for the bufadienolides (Enzo Life Sciences, Farmingdale, NY) and prostaglandins (Cayman Chemicals, Ann Arbor, MI). The final assay mixture (100 µL) contained 5 µM Cgr2, 50 µM flavin, 0.375 mM MV, 0.25 mM dithionite, and was initiated by addition of 0.5 mM substrate. For metal activation studies, metal salts were dissolved in pH 7 buffer (1 mM) and added to a final concentration of 40 µM. Assays were prepared in a 96-well polysterene microplate (Corning, Corning, NY) and activity was continuously monitored by measuring the absorbance at 600 nm on a PowerWave HT Microplate Spectrophotometer (BioTek, Winooski, VT); a decrease in the absorbance at 600 nm corresponded to MV oxidation coupled to substrate reduction. For endpoint assay, reactions were quenched in methanol, diluted to a final concentration of 1 µM in 50% methanol, and analyzed by LC-MS/MS as previously described.

## Kinetic assays

Kinetic assays were performed in an anaerobic glovebox (Mbraun, Stratham, NH) at 25°C. Reactions were run in triplicate (200 µL) in assay buffer containing 5 µM Cgr2, 500 µM FAD, 1.5 mM MV, and 1 mM sodium dithionite, and were initiated by addition of digoxin as a solution in DMF to a final concentration of 0.01, 0.025, 0.05, 0.1, 0.2, 0.25, 0.3, and 0.5 mM. 20 µL reaction aliquots were quenched in 180 µL of ice-cold methanol in Costar flat bottom polysteryene 96-well plates (Corning, Corning, NY). The plates were sealed with adhesive aluminum foil for 96-well plates (VWR, Radnor, PA), brought out of the anaerobic chamber, and further diluted (50-fold) into 50% methanol. Digoxin and dihydrodigoxin standard curves were prepared in the full reaction matrix and processed identically such that final concentrations (after 500x total dilution) generated a standard curve between 0.01–1 µM. Plates were centrifuged (4000 rpm x 10 min, 4°C) and 200 µL of each reaction were transferred to a 0.5 mL PP 96-well plate (Agilent Technologies, Santa Clara, CA) sealed with EPS easy piercing seals (BioChromato, San Diego, CA). The reactions were monitored by LC-MS/MS as previously described, except that samples were directly injected (no column), and isocratic flow was used with 75% methanol with 1 mM ammonium hydroxide.

## Chemical similarity analysis

The chemical similarity of all substrates was assessed using the ChemMine software (http://chemminetools.ucr.edu) (*Backman et al., 2011*). Substrates were imported into ChemMine in SMILES format. The hierarchical clustering tool was used to generate a heatmap visualizing the structural distance matrix between each substrate and digoxin.

## Cgr2 sequence analysis

The full length Cgr2 protein sequence from *E. lenta* DSM 2243 was used as a query for BLASTP (*Atlschul et al., 1997*) using the NCBI non-redundant protein sequence database (search performed

9/26/17). Cgr2 was also used to query the HHPred prediction tool (https://toolkit.tuebingen.mpg.de/#/tools/hhpred) to identify additional remote protein homologs using hidden Markov models (*Alva et al., 2016*). The PDB_mmCIF70_27_Aug database was used (search performed 9/26/17).

## Construction of sequence similarity network (SSN)

A SSN was generated using the EFI-EST tool (http://efi.igb.illinois.edu/efi-est/) (*Gerlt et al., 2015*). The full length (native) Cgr2 protein sequence was used as an input to generate a network with the 5000 most similar sequences from the UniProtKB protein database. An initial alignment score cutoff of $10^{-66}$ generated a SSN with 2018 nodes (with 100% identity) and 317,130 edges. The SSN was imported into Cytoscope v 3.2.1 and visualized with the 'Organic layout' setting. Seven characterized enzymes were present within the network (UniProtKB IDs: fumarate reductases: P83223, P0C278, Q07WU7, Q9Z4P0; urocanate reductase: Q8CVD0; 3-oxosteroid-1-dehydrogenase: P71864; Q7D5C1). The alignment score cutoff was increased to e-value $<10^{-130}$, until enzymes with known functions separated into putatively isofunctional clusters. At this threshold, Cgr2 appears as a singleton. The network shown in *Figure 5A* was generated with a cutoff of e-value $<10^{-50}$, a threshold at which nearly all protein sequences form one cluster. Multiple sequence alignments were generated in Geneious and visualized in Jalview (clustalx coloring). To validate that the clusters in the SSN likely contained isofunctional proteins, Cgr2 was aligned with characterized enzymes and additional selected proteins within the corresponding clusters of the SSN, and the alignment was analyzed for the presence of conserved active site residues involved in substrate binding, activation and proton transfer (*Leys et al., 1999*; *Rohman et al., 2013*; *Bogachev et al., 2012*; *Knol et al., 2008*; *Reid et al., 2000*).

## *E. lenta* and *cgr2* abundance and prevalence

*E. lenta* and *cgr2* prevalence were determined using the copy number abundance (gene copies/cell) as derived from Metaquery2 (*Nayfach et al., 2015*) using the median abundance from individuals with repeated sampling. *E. lenta* abundance was determined from a single copy *E. lenta* marker gene described elsewhere (*elnmrk1*) (*Bisanz et al., 2018*). Matches were required to have a minimum 90% nucleotide identity and query/target coverage. Reconstruction of metagenomic *cgr2* sequences was carried out by quality trimming reads from 96 metagenomes with >0.001 proportional abundance of *E. lenta* or >1 fold coverage using default sliding window settings with Trimmomatic (*Bolger et al., 2014*) and extracting reads which mapped to the *cgr* cluster and associated intergenic space (2957889..2968387) in the reference DSM 2243 assembly with Bowtie 2. These were assembled and annotated as above. Alignments to Cgr2 in metagenomic coding sequences were filtered by a global alignment identity of >80% to position 333 ± 60 residues. For assembly-free variant calling, reads were filtered for a minimum mapping quality of 10 and a pileup was created (SAMtools). 49 metagenomes had at least one read mapping to the variant position (2959294). Variants were called when > 50% of reads at a site supported an alternative sequence from the reference. Conservation of nucleotide sequence in isolates was independently confirmed via Sanger sequencing (GENEWIZ, San Francisco, CA, USA) using the following primers: cgr2_fwd (TGCAATCAAGACAACCACGA), cgr2_internal (TCGGTGTACAACCACAATGC), and cgr2_rev (GTTGCGCTGTGATTAGACTG). PCR was carried out with high-fidelity Q5 enzyme (New England Biolabs, Ipswich, MA).

To validate metagenomics inquiries, qPCR analysis with double-dye probes was carried out in a duplexed fashion using the following primers and probes: ElentaUni_F (GTACAACATGCTCCTTGCGG), ElentaUni_R (CGAACAGAGGATCGGGATGG), ElentaUni_Probe ([6FAM]TTCTGGCTGCACCGTTCGCGGTCCA[BHQ1]), cgr2_F (GAGGCCGTCGATTGGATGAT), cgr2_R (ACCGTAGGCATTGTGGTTGT), and cgr2_probe ([HEX]CGACACGGAGGCCGATGTCG[BHQ1]). Reactions were carried out in triplicate using 10 µL reactions with 200 nM primers and probes using BioRad Universal Probes Supermix on a BioRad CFX 384 thermocycler according to the manufacturer's suggested settings for fast cycles with a 60 °C annealing temperature. The estimated assay detection limit based on spike-in experiments is $1.4 \times 10^3$ GE/g after accounting for DNA extraction. Human samples were collected for the purpose of microbiome analysis as part of the following registered studies: NCT03022682, NCT01967563, and NCT01105143 and approved by their respective institutional review boards. DNA was extracted with variable methods using either MoBio Power Soil

(QIAGEN, Germantown, MD), Qiagen Fast Stool (QIAGEN, Germantown, MD), or Promega Wizard (Promega, Madison, WI) SV 96 kits.

## Statistical analysis

All statistical analysis was carried out using either Student's t-test as implemented in Graphpad Prism version 7 (La Jolla, CA, USA) or R version 3.4.0 using appropriate base functions for Welch's t-test, Pearson and Spearman correlations, and ANOVA with multcomp version 1.4–6 for Dunnett's multiple comparison test. Graphing was carried out with Graphpad Prism and R using ggplot2 version 2.2.1. Skewedness was calculated using the R package Moments version 0.14.

## Acknowledgements

We thank the National Institute of Advanced Industrial Science and Technology (Tokyo, Japan) for providing the *Rhodococcous erythropolis* L-88 strain and pTipQ expression vectors, José Miguel Sahuquillo-Arce (Hospital Universitari i Politecnic La Fe, Spain) for providing the *Eggerthella lenta* Valencia strain, and Emma Allen-Vercoe (University of Guelph, Canada) for isolating and supplying additional strains of *Eggerthella lenta.* We are indebted to Patrick Bradley, Stephen Nayfach, Katherine Pollard, and Jack Nicoludis for pivotal discussions; and to Lauren Rajakovich for comments on the manuscript. We are also indebted to Stephen Nayfach for supplying metagenomes from the Metaquery2 set for further analysis. This work was supported by the National Institutes of Health (PJT: R01HL122593; MP: GM111978), the Searle Scholars Program (EB:12-SSP-243; PJT:SSP-2016– 1352), a Fellowship for Science and Engineering from the David and Lucille Packard Foundation (EB:2013–39267), the George W. Merck Fellowship (EB:27–14), the Bill and Melinda Gates Foundation (EB:OPP1158186), and the UCSF Department of Microbiology and Immunology (PJT). PJT is a Chan Zuckerberg Biohub investigator and a Nadia's Gift Foundation Innovator supported, in part, by the Damon Runyon Cancer Research Foundation (DRR-42–16). PJT is also supported by the UCSF Program for Breakthrough Biomedical Research (partially funded by the Sandler Foundation). JEB is a Natural Sciences and Engineering Research Council of Canada postdoctoral fellow. NK received funding from the National Institutes of Health (GM095450-01), the Smith Family Graduate Science and Engineering Fellowship, and a National Science Foundation Graduate Research Fellowship (DGE1144152).

## Additional information

### Competing interests

Peter J Turnbaugh: PJT is on the scientific advisory board for Seres Therapeutics, WholeBiome, and Kaleido, and has active research funding from Medimmune. In the past year PJT has consulted for GLG and MEDAcorp. Emily P Balskus: EPB is a consultant for Merck Research Labs, Novartis, and Kintai Therapeutics. The other authors declare that no competing interests exist.

### Funding

| Funder | Grant reference number | Author |
|---|---|---|
| Smith family | Graduate Science and Engineering Fellowship | Nitzan Koppel |
| National Science Foundation | Graduate student fellowship, DGE1144152 | Nitzan Koppel |
| National Institutes of Health | Training grant, GM095450-01 | Nitzan Koppel |
| Natural Sciences and Engineering Research Council of Canada | Postdoctoral fellowship | Jordan E Bisanz |
| National Institutes of Health | GM111978 | Maria-Eirini Pandelia |
| National Institutes of Health | R01HL122593 | Peter Turnbaugh |

| Searle Scholars Program | SSP-2016-1352; EB:12-SSP-243 | Peter Turnbaugh |
| UCSF Department of Microbiology and Immunology | | Peter Turnbaugh |
| Damon Runyon Cancer Research Foundation | DRR-42-16 | Peter Turnbaugh |
| Chan Zuckerberg Biohub | | Peter Turnbaugh |
| University of California, San Francisco | Program for Breakthrough Biomedical Research | Peter Turnbaugh |
| David and Lucile Packard Foundation | 2013-39267 | Emily P Balskus |
| George W. Merck Fellowship | 27-14 | Emily P Balskus |
| Bill and Melinda Gates Foundation | OPP1158186 | Emily P Balskus |
| Searle Scholars Program | 12-SSP-243 | Emily P Balskus |

The funders had no role in study design, data collection and interpretation, or the decision to submit the work for publication.

## Author contributions

Nitzan Koppel, Conceptualization, Formal analysis, Investigation, Methodology, Writing—original draft, Writing—review and editing; Jordan E Bisanz, Data curation, Formal analysis, Investigation, Visualization, Writing—original draft, Writing—review and editing; Maria-Eirini Pandelia, Formal analysis, Investigation, Methodology, Writing—review and editing; Peter J Turnbaugh, Conceptualization, Resources, Formal analysis, Supervision, Funding acquisition, Investigation, Methodology, Writing—original draft, Project administration, Writing—review and editing; Emily P Balskus, Conceptualization, Resources, Supervision, Funding acquisition, Writing—original draft, Project administration, Writing—review and editing

## Author ORCIDs

Nitzan Koppel [iD] http://orcid.org/0000-0002-8399-2943
Jordan E Bisanz [iD] http://orcid.org/0000-0002-8649-1706
Maria-Eirini Pandelia [iD] http://orcid.org/0000-0002-6750-1948
Peter J Turnbaugh [iD] https://orcid.org/0000-0002-0888-2875
Emily P Balskus [iD] https://orcid.org/0000-0001-5985-5714

## Ethics

Human subjects: Human samples were collected for the purpose of microbiome analysis as part of the following registered studies: NCT03022682, NCT01967563, and NCT01105143 and approved by their respective institutional review boards.

## Decision letter and Author response

Decision letter https://doi.org/10.7554/eLife.33953.043
Author response https://doi.org/10.7554/eLife.33953.044

# Additional files

## Supplementary files

• Transparent reporting form
DOI: https://doi.org/10.7554/eLife.33953.038

## Data availability

Data used was taken from http://metaquery.docpollard.org/ (job ids: 703, 805, 807, 808-813).

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
