## [Decision Letter]

Thank you for submitting your article "Discovery and characterization of a prevalent human gut bacterial enzyme sufficient for the inactivation of plant toxins" for consideration by *eLife*. Your article has been reviewed by three peer reviewers, and the evaluation has been overseen by a Reviewing Editor and Wendy Garrett as the Senior Editor. The reviewers have opted to remain anonymous.

The reviewers have discussed the reviews with one another and the Reviewing Editor has drafted this decision to help you prepare a revised submission.

Summary:

This report extends previous work by the Balkus and Turnbaugh groups that linked digoxin metabolism to a unique two-gene locus within *Eggerthella lenta*. Here the authors demonstrate that Cgr2 is a novel reductase that requires one or more [4Fe-4S] clusters and FAD+ for the specific conversion of the cardenolide digoxin and closely related compounds. Using site-directed mutagenesis, the authors narrow down which Cys are likely involved in cluster formation and compare the activity of two variants of the enzyme (N333 and Y333) detected within different *E. lenta* genomes. The significance of these two variants is unknown but relevant in terms of digoxin metabolism by *E. lenta*. Finally, the authors show via sequence comparisons with known reductases that Cgr2 represents a novel family and that its presence is widespread in the human gut microbiota, which underscores the overall importance of these data in understanding discrete xenobiotic transformations by this community.

The three reviewers find the study interesting and that is presents a valuable example for discovering the mechanistic basis of xenometabolic potential of the members of the human gut microbiome. There are some points that should be addressed in a revised version:

Essential revisions:

One important unresolved question is whether this enzyme carries out plant toxin detoxification function in vivo. This should be discussed in more detail; as written now the paper leaves the impression that the presence of these enzymes in the gut is enough to assume function. The authors suggest in the introduction and discussion that cgr2 is akin to intestinal and hepatic enzymes that are rapid responders to toxins. The possibility that a very low abundance organism like *E. lenta* and a very low abundance gene like cgr2 are acting as specialized detoxification systems in the gut is fascinating but not directly tested in this work. Is it not possible to do ex vivo studies on digoxin reduction in fecal samples with *cgr*+/*cgr- E. lenta* carriers given that qPCR data suggests many people are carrying *E. lenta*? How would one discount the possibility that enzymes other than cgr2 in the gut are turning over digoxin?

A major argument the authors make is the cgr2 gene is widespread and conserved in the human microbiome; they also note that, while present, *E. lenta* is generally very low abundance. Their analysis of genomes and qPCR data indicates that cgr2 is not present in every *E. lenta* genome. This makes sense in light of their observation in cultured isolates that the whole cgr cassette is gained and lost. Please expand on how the gain and loss and the flexible carriage of this gene in *E. lenta* dovetails with the assertion that the gut microbiome is maintaining the gene specifically to deal with plant toxins.

What is the role/function of the additional 6 gene cac locus that co-occurs with Cgr? Is it important since it does not turn on as Cgr does with digoxin? Is Cgr1 important for digoxin metabolism? (The authors speculate that Cgr1 aids in electron transfer, though lack of expression in *Rhodococcus* still permitted digoxin reduction, perhaps due to endogenous factors that compensate in this bacterium.) A discussion on the potential function of the other genes in the conserved cluster would be helpful in getting a more all-round view on the findings.

It is puzzling that the identified gene cluster is conserved – across isolates spanning 3 continents and over 70 years – at the nucleotide level identity of 99.95 ± 0.05% (only 5 SNVs on average and cases where there are fewer or none). This is highly unlikely. It is hard to imagine any biological reason that leaves no room even for synonymous substitutions. This finding needs to be carefully rechecked/explained and additional analysis and controls based on, e.g., other genomic regions need to be provided.

---

## [Author Response]

Essential revisions:One important unresolved question is whether this enzyme carries out plant toxin detoxification function in vivo. This should be discussed in more detail; as written now the paper leaves the impression that the presence of these enzymes in the gut is enough to assume function. The authors suggest in the introduction and discussion that cgr2 is akin to intestinal and hepatic enzymes that are rapid responders to toxins. The possibility that a very low abundance organism like E. lenta and a very low abundance gene like cgr2 are acting as specialized detoxification systems in the gut is fascinating but not directly tested in this work. Is it not possible to do ex vivo studies on digoxin reduction in fecal samples with cgr+/cgr- E. lenta carriers given that qPCR data suggests many people are carrying E. lenta?

We agree with the reviewers that this is an important question, and we have performed the proposed experiments in a previous study (Haiser et al., 2013). Stool samples from 20 unrelated healthy individuals were analyzed by quantitative PCR to determine the *cgr* ratio (*cgr* operon/*E. lenta* 16S rRNA gene copies) and were also subjected to ex vivo incubations with digoxin. Gut microbial communities with a high *cgr* ratio metabolized digoxin to a much greater extent (96.25% reduction) compared to microbiomes with low *cgr* ratios (12.8% reduction). The baseline abundance of the *E. lenta* species did not show any significant correlation with digoxin reduction. Together, these results provided initial evidence that individuals enriched for *cgr*+ *E. lenta* (even in low abundance) have a higher capacity to metabolize digoxin. Furthermore, in this previous paper, germ-free mice that had been mono-colonized by *cgr*+ *E. lenta* had lower serum levels of digoxin than mice that were colonized by *cgr- E. lenta* when fed a protein-free diet. Together, these experiments demonstrate that metabolism of digoxin by Cgr2 is relevant in vivo.

To more clearly highlight this important previous work, we have clarified the results and details of these experiments in the Introduction:

“The cgr operon was absent in two *E. lenta* strains that did not metabolize digoxin (“non-reducing” strains) and *cgr* operon presence and abundance predicted the extent of drug inactivation by human gut microbial communities in ex vivo incubations (8). Furthermore, germ-free mice that had been mono-colonized by a reducing (*cgr*+) strain of *E. lenta* had lower serum levels of digoxin than mice colonized by a non-reducing (*cgr*-) strain, and dietary arginine efficiently blocked digoxin reduction by the cgr operon in *cgr*+ *E. lenta*-colonized mice (8, 9).”

How would one discount the possibility that enzymes other than cgr2 in the gut are turning over digoxin?

In the seminal studies on digoxin metabolism by John Lindenbaum and colleagues (Columbia University), researchers screened >400 gut bacterial isolates from two cardiac patients that had been previously shown to excrete high levels of dihydrodigoxin (referred to in these studies as digoxin reduction products (DRP), (Dobkin et al., 1983)). Only two colonies produced detectable DRP, and these organisms were identified as *Eggerthella lenta* (referred to at the time as *Eubacterium lentum*) using biochemical and chromatographic methods. A follow-up screen of 150 stock strains also only found *E. lenta*, while also highlighting that digoxin reduction is a strain-variable trait (18/28 tested strains produced DRP). To date, no other microbes, including closely related *Coriobacteriia* species, have been shown to possess this metabolic activity. Although we cannot rule out the possibility that additional as-of-yet undiscovered enzymes and microbes may be involved in digoxin reduction, this previous finding and the study mentioned in the above response are both consistent with the hypothesis that *cgr*+ *E. lenta* are primarily responsible for this metabolic activity in the human gut microbiome.

We have now incorporated a description of these previous findings into the Introduction:

“It has been known for decades that human gut bacteria reduce digoxin to the inactive metabolite dihydrodigoxin, decreasing drug efficacy and toxicity (3-5). Screening hundreds of gut bacterial strains from humans that excreted high levels of dihydrodigoxin revealed only two isolates that were capable of metabolizing digoxin, both of which were strains of the anaerobic, low abundance bacterium *Eggerthella lenta (3*).”

We also explicitly mentioned that we cannot rule out the possibility that additional, undiscovered gut enzymes mediate this transformation in the Discussion section:

“Although we cannot rule out the possibility that other enzymes are involved in digoxin inactivation, no other microbes have been discovered that possess this metabolic activity apart from *cgr2+ E. lenta (3*).”

A major argument the authors make is the cgr2 gene is widespread and conserved in the human microbiome; they also note that, while present, E. lenta is generally very low abundance. Their analysis of genomes and qPCR data indicates that cgr2 is not present in every E. lenta genome. This makes sense in light of their observation in cultured isolates that the whole cgr cassette is gained and lost. Please expand on how the gain and loss and the flexible carriage of this gene in E. lenta dovetails with the assertion that the gut microbiome is maintaining the gene specifically to deal with plant toxins.

Our genomic and metagenomic analysis expand on the initial observation that the genes necessary for the metabolism of cardenolides are found within the variable portion of the *E. lenta* pan-genome. A clear implication of these results is that *cgr2* is non-essential for the fitness of all *E. lenta* strains within the gastrointestinal tract, consistent with the lack of a clear endogenous substrate or either an in vitro or in vivo growth phenotype.

Incubation of both *cgr*+ and *cgr-* strains with digoxin during growth in rich media did not show any significant change in growth rate or carrying capacity (Haiser et al., 2013).

Similarly, we also did not see a significant shift in colonization level in germ-free mice mono-associated with *cgr*+ and *cgr-* strains (Haiser et al., 2013).

One potential reason for the lack of a growth advantage for *cgr*+ strains is that other electron acceptors are present in rich medium (BHI) and in the gastrointestinal tract. To address this point, we have now included an additional experiment, wherein we demonstrate that *E. lenta* cannot use digoxin or other cardenolides as a sole electron acceptor during in vitro growth in minimal media (Figure 4—figure supplement 2). DMSO is used as a positive control to show that *E. lenta* is capable of anaerobic respiration. These results are now discussed in the main text (subsection “Cgr2 is a novel enzyme that preferentially reduces cardenolides”):

“Additionally, neither fumarate nor any of the metabolized cardenolides conferred a growth advantage to *cgr2+ E. lenta* in minimal or rich medias, suggesting that these compounds are not used as alternative terminal electron acceptors (Figure 4—figure supplement 2).”

Taken together, these studies support the surprising hypothesis that *cgr2* may have evolved to benefit the host by protecting against plant toxins, analogous to other intestinal enzymes for drug metabolism (e.g., CYP450s).

We have expanded these points in the revised Discussion section:

“The high sequence conservation and levels of *cgr* operon transcription in response to digoxin exposure suggest that digoxin metabolism may provide a physiological benefit to *E. lenta*. However, as we could not observe any direct benefit of cardenolide metabolism for *cgr*+ *E. lenta* (Figure 4—figure supplement 2), we hypothesize that these bacteria may have evolved to protect the host against plant toxins and thus maintain a habitat for colonization.”

What is the role/function of the additional 6 gene cac locus that co-occurs with Cgr? Is it important since it does not turn on as Cgr does with digoxin?

The co-occurrence and high sequence conservation of the *cac* genes and the *cgr* operon is potentially suggestive of a role for some (or all) of the *cac* genes in cardenolide metabolism. As the reviewer astutely points out, none of the *cac* genes were significantly up-regulated in response to digoxin, although *cac3* trended towards increased expression (1.5-fold relative to vehicle controls; see Figure 1E).

The lack of transcriptional induction does not rule out a role for these genes in cardenolide metabolism, as they could be constitutively expressed or post-transcriptionally regulated. To look into this in more detail, we analyze the expression of this genomic locus. In both the studied dataset (Author response image 1), and in an unpublished dataset (not shown), the remainder of the *cgr-*associated gene cluster is largely transcriptionally dormant (or expressed below our limit of detection) under varying conditions of arginine concentration, growth phase, and digoxin treatment.

**Author response image 1. respfig1:** RNA-Seq read depth across the *cgr-*associated gene cluster and neighboring regions. While the *cgr*-operon is induced in the presence of digoxin in exponential phase and stationary phase under low arginine conditions, the remainder of the cluster is relatively transcriptionally dormant, with the exception of a small degree of transcription of the *cac4* reductase independent of digoxin in stationary phase.

Is Cgr1 important for digoxin metabolism? (The authors speculate that Cgr1 aids in electron transfer, though lack of expression in Rhodococcus still permitted digoxin reduction, perhaps due to endogenous factors that compensate in this bacterium.) A discussion on the potential function of the other genes in the conserved cluster would be helpful in getting a more all-round view on the findings.

Our working hypothesis is that Cgr1 is important for digoxin metabolism in vivo, but serves a more general role, such as electron transfer to Cgr2 or localization to the *E. lenta* membrane, rather than participating directly in substrate binding and catalysis. We identified a close homolog of Cgr1 (>91% amino acid identity) in both metabolizing and non-metabolizing strains of *E. lenta*, which supports the hypothesis that Cgr1 is involved in a more general function rather than direct reduction of digoxin. These proposed roles would be consistent with the roles of biochemically characterized Cgr1 homologs (NrfH from *Desulfovibrio vulgaris* and *Wolinella succinogenes* and CymA from *Shewanella putrefaciens*), which participate in transferring electrons to associated partner reductase enzymes. The genes encoding the Nrf system are organized in a similar way to the *cgr* operon, with a reductase gene adjacent to its cytochrome *c* reductase partner (Simon et al., 2000). Elucidating the specific function of Cgr1 would require the use of alternative heterologous expression systems (with the appropriate cytochrome *c* maturation systems to install covalent heme groups) or the development of genetic tools in *E. lenta*, all of which are beyond the scope of this study.

We have now included a discussion of the potential role of Cgr1 in digoxin metabolism into the Results section:

“We also identified a close homolog of Cgr1 (Elen_2528) in *E. lenta* DSM 2243 (91.75% amino acid identity, BLASTP) that is a component of the *E. lenta* core genome (99.39 ± 0.81% global identity mean ± SD). The presence of this highly similar protein in both metabolizing and non-metabolizing strains further indicates that Cgr1 is involved in a more general function (e.g. electron transfer, membrane docking) rather than direct reduction of digoxin.”

We also mention potential future directions in the revised Discussion section:

“While we have demonstrated that Cgr2 is necessary and sufficient for digoxin reduction in a heterologous host and in vitro using a chemical electron donor, additional proteins within the expanded cgr gene cluster may be important for digoxin activity in vivo. Sequence analyses and transcriptional data suggest that Cgr1 is likely important for this metabolic activity in *E. lenta*. However, we were unable to observe overexpression or heme *c* incorporation into Cgr1 using a variety of heterologous constructs, hosts, and expression conditions, which may be due to an incompatibility of heterologous cytochrome *c* maturation factors and this protein (59, 60). The use of alternative heterologous systems that are more suitable for producing multi-heme cytochromes *c* (61, 62) or the development of genetic tools in *E. lenta* would thus be required to obtain functional Cgr1 and determine its role in digoxin metabolism.”

It is puzzling that the identified gene cluster is conserved – across isolates spanning 3 continents and over 70 years – at the nucleotide level identity of 99.95 ± 0.05% (only 5 SNVs on average and cases where there are fewer or none). This is highly unlikely. It is hard to imagine any biological reason that leaves no room even for synonymous substitutions. This finding needs to be carefully rechecked/explained and additional analysis and controls based on, e.g., other genomic regions need to be provided.

We wholeheartedly agree – the degree of conservation at the nucleotide level may be viewed as surprising, especially for a gene that is not found in the core portion of the *E. lenta* pan-genome. To rule out potential assembly artifacts, we had originally confirmed the sequence of *cgr2* in the entire strain collection by Sanger sequencing (briefly described in the Materials and methods section of our original submission). Sequencing from both ends of *cgr2*, and with an internal read, we assembled the *cgr2* gene by direct overlap (CAP3) and then carried out multiple alignment with Clustal Omega. The result is shown below confirming observations found during genome analysis (Author response image 2).

**Author response image 2. respfig2:** Clustal Omega alignment of Sanger-sequenced *cgr2* confirms high degree of conservation.

Having validated the accuracy of our genome assemblies on *cgr2*, we extended the analysis to the other strains applying an assembly free method. Briefly, reads were mapped to the DSM 2243 reference assembly (Bowtie 2), filtered for high quality unique mappings (>=mapping quality 10) and pileups were created for variant calling (Samtools view and mpileup). This confirmed our results in *cgr2* as well as the degree of conservation in the remainder of the *cgr*-associated gene cluster (Author response image 3, also shown in revised Figure 3B). As previously reported in the original submission, all 7 additional strains of *cgr2*+ *E. lenta* show a M>V variant at position 381 and 5/8 strains show the Y>N variant at position 333. Limited variation is observed elsewhere with the exception of within the non-essential *cgr1* gene of *E. lenta* 11C (El2) where 13 variants are found leading to a 98% global nucleotide identity.

**Author response image 3. respfig3:** Variant calling within the *cgr*-associated gene cluster. Mapping of reads to the reference assembly and calling of variants confirms assembly-based analysis wherein an average of 4.14 variants are called in the cluster (median = 3, range 2-14).

Next, we sought to determine where *cgr2* and the other associated genes fall within the rest of the *E. lenta* pan-genome, i.e. are they extreme outliers or are many genes equally conserved? Surprisingly, *cgr2* is only at the 67^th^ percentile of amino acid conservation in the pan-genome (78.8^th^ in the core genome, and 58.5^th^ in the non-singleton accessory genome). This finding is illustrated in our new Figure 3C.

This analysis was similarly run over metagenomes and incorporated into the revised Figure 3. We have now included an in-depth discussion of these validation efforts and new analyses within the Results section:

“To identify additional amino acids that may be important for Cgr2 function, we compared the Cgr2 sequences encoded within our collection of *E. lenta* genomes. Strikingly, only two *cgr2* nucleotide variants were detected which were validated by targeted Sanger sequencing. One of these variants is only found in the DSM 2243 type strain resulting in a conservative methionine (M) to valine (V) substitution at position 381. The other results in either aromatic tyrosine (Y) as in the type strain DSM 2243 or neutral asparagine (N) at position 333 (Figure 3A). We were also able to fully or partially reconstruct 14 additional *cgr2* sequences using reads mapping to the *cgr* gene cluster from 96 gut microbiome datasets with a high abundance of *E. lenta* (> 1x coverage or > 0.001 proportional abundance). These metagenome fragments confirmed the presence of both Y333 and N333 variants in a 9:5 ratio (Figure 3A) while the DSM 2243 M381 variant was not observed. To avoid biases against lower *E. lenta* coverage metagenomes, we also applied an assembly-free method based on calling variants from aligned reads (Figure 3B). This uncovered 49 metagenomes with at least one read mapping over the variant position confirming the bi-allelic nature with 15 Y333 and 34 N333 metagenomes. Nearly all metagenomes (41/42) with reads mapping to position 381 supported the valine residue suggesting the DSM 2243 M381 variant is rare. Given that this analysis confirmed the highly conserved nature of the *cgr* locus, we analyzed the conservation of *cgr2* in the context of the *E. lenta* pan-genome (based on 24 sequenced isolates) finding that it is at the 67th percentile of conservation. These results suggest that *cgr2* sequence conservation is not unusual for this species, with the caveat that relatively few genomes were available for analysis (Figure 3C).”

We have also revised the Discussion section to address these points and the broader question of the drivers of bacterial genome evolution:

“These results, together with the overall high degree of conservation in the *E. lenta* pan-genome, suggest that much of the phenotypic variation within this species may be driven by gene gain/loss rather than by genetic polymorphisms. Data from other bacterial lineages suggests that this phenomenon may not be unique to *Coriobacteriia* (68-72); in *Pseudomonas syringae*, 1% amino acid divergence accumulates at the same time in hundreds or even thousands of genes (71).”